# Bile salt receptor complex activates a pathogenic type III secretion system

Peng Li[1†], Giomar Rivera-Cancel[1†], Lisa N Kinch[2], Dor Salomon[1], Diana R Tomchick[3,4], Nick V Grishin[2,3,4], Kim Orth[1,2,3*]

[1]Department of Molecular Biology, University of Texas Southwestern Medical Center, Dallas, United States; [2]Howard Hughes Medical Institute, University of Texas Southwestern Medical Center, Dallas, United States; [3]Department of Biochemistry, University of Texas Southwestern Medical Center, Dallas, United States; [4]Department of Biophysics, University of Texas Southwestern Medical Center, Dallas, United States

**Abstract** Bile is an important component of the human gastrointestinal tract with an essential role in food absorption and antimicrobial activities. Enteric bacterial pathogens have developed strategies to sense bile as an environmental cue to regulate virulence genes during infection. We discovered that *Vibrio parahaemolyticus* VtrC, along with VtrA and VtrB, are required for activating the virulence type III secretion system 2 in response to bile salts. The VtrA/VtrC complex activates VtrB in the presence of bile salts. The crystal structure of the periplasmic domains of the VtrA/VtrC heterodimer reveals a β-barrel with a hydrophobic inner chamber. A co-crystal structure of VtrA/VtrC with bile salt, along with biophysical and mutational analysis, demonstrates that the hydrophobic chamber binds bile salts and activates the virulence network. As part of a family of conserved signaling receptors, VtrA/VtrC provides structural and functional insights into the evolutionarily conserved mechanism used by bacteria to sense their environment.

*For correspondence: kim.orth@utsouthwestern.edu

[†]These authors contributed equally to this work

Competing interests: The authors declare that no competing interests exist.

## Introduction

Bile is an important component of the human gastrointestinal tract that plays a major role in the emulsification and solubilization of lipids (*Begley et al., 2005*). Its main components are bile salts, cholesterol, and phospholipids that together possess antimicrobial activities, challenging the survival and colonization of both pathogenic and commensal bacteria (*Begley et al., 2005*). There is increasing evidence that enteric bacterial pathogens can utilize bile as a signaling cue to regulate gene expression profiles during host infection. For example, *Salmonella typhimurium* senses bile to repress the genes involved in invasion when growing in the intestinal lumen and induce their expression to promote invasion upon the penetration of the mucosal layer wherein the concentration of bile is decreased (*Prouty and Gunn, 2000*; *Prouty et al., 2004*). In *Shigella spp.*, bile salts increase the pathogen's adherence and invasion of epithelial cells (*Pope et al., 1995*). Bile also has important effects on the pathogenicity of *Vibrio* species, as observed in multiple studies with pathogenic strains of *V. cholerae* and *V. parahaemolyticus* (*Gupta and Chowdhury, 1997*; *Schuhmacher and Klose, 1999*; *Krukonis and DiRita, 2003*; *Faruque et al., 1998*). *V. cholerae* produces two major virulence factors during infection, cholera toxin (CT) and toxin-coregulated pilus (TCP), and these factors are repressed by ToxT in the presence of bile (*Gupta and Chowdhury, 1997*; *Schuhmacher and Klose, 1999*; *Krukonis and DiRita, 2003*; *Faruque et al., 1998*). However, it was also reported that bile can activate the production of CT independent of ToxT (*Hung and Mekalanos, 2005*). Non-O1/non-O139 *V. cholerae* strains that do not encode CT and TCP can cause gastroenteritis and utilize bile to activate a pathogenic type III secretion system (*Dziejman et al., 2005*;

---

**eLife digest** When we eat a meal, bile salts in the intestine help the body to absorb molecules of fat. Certain bacteria that cause food poisoning in humans, such as *Vibrio parahaemolyticus*, can also detect bile salts and use them to sense that they have reached the intestine of a suitable host. The bacteria then produce toxins that inflame the intestine; this can result in days of diarrhea. However, it was not clear how the bacteria detect the bile salt signal and how this triggers them to produce the toxins.

Li, Rivera-Cancel et al. now discover that not one, but two, genes are required for this process. The genes encode two proteins, called VtrA and VtrC, that interact to form protein 'complex' on the surface of the membrane that surrounds the bacterial cell. The two proteins create a barrel-like structure that can bind to bile salts and trigger the cell to produce the toxins.

Future experiments will aim to understand how the binding of bile salts to this protein complex causes an increase toxin production. A future challenge is to find out how other disease-causing bacteria sense environmental cues to produce toxins. With this knowledge, researchers might be able to design new drugs that could prevent the production of toxins to relieve symptoms of food poisoning and other illnesses.

---

*Chaand et al., 2015*; *Alam et al., 2010*). Similarly, *V. parahaemolyticus* type III secretion system 2 (T3SS2) is induced specifically by bile salts during infection, resulting in acute gastroenteritis (*Gotoh et al., 2010*). Despite these important discoveries, the mechanism of sensing bile salts by pathogenic bacteria remains unknown. To investigate this mystery, we used *V. parahaemolyticus* as a model to elucidate how *Vibrio spp.* sense bile salts as a signal to regulate the expression of virulence genes.

*V. parahaemolyticus* is a globally-spread, Gram-negative, halophilic bacterial pathogen that inhabits marine and estuarine environments and is the world's leading cause of acute gastroenteritis due to the consumption of raw or undercooked seafood (*Broberg et al., 2011*; *Zhang and Orth, 2013*). During recent years, rising temperatures in the ocean has contributed to this pathogen's worldwide dissemination (*Nair et al., 2007*; *Velazquez-Roman et al., 2013*; *McLaughlin et al., 2005*; *O'Boyle and Boyd, 2014*; *Daniels et al., 2000*). *V. parahaemolyticus* is also the causative agent of the devastating shrimp disease Acute Hepatopancreatic Necrosis Disease (AHPND) (*Tran et al., 2013*; *Lee et al., 2015*).

Over the last decade, enormous progress has been made to elucidate virulence factors used by this pathogen and tools are available to study virulence mechanisms at the genetic and biochemical levels (*de Souza Santos et al., 2015*). As part of its virulence repertoire, *V. parahaemolyticus* encodes two T3SSs: the cytotoxic T3SS1 and the enterotoxic T3SS2. T3SSs encode needle-like secretion apparatuses used to deliver bacterial effector proteins, called Vops, which allow *V. parahaemolyticus* to invade and kill infected host cells (*Broberg et al., 2011*; *Makino et al., 2003*; *Galán and Wolf-Watz, 2006*). Each T3SS delivers a specific set of Vops into a host cell (*Broberg et al., 2011*). For example, the T3SS1 exclusively secretes VopQ, VopR, VPA0450 and VopS and uses the effectors to orchestrate a multifaceted host cell death. This system is found in all strains of *V. parahaemolyticus*, and therefore is proposed to be important for bacterial survival in the environment (*Broberg et al., 2011*).

T3SS2 is present in clinical isolates of *V. parahaemolyticus* and is a key bile salt-induced virulence system that causes enterotoxicity and symptoms associated with gastroenteritis during infection (*Broberg et al., 2011*; *Park et al., 2004*; *Hiyoshi et al., 2010*; *Ritchie et al., 2012*). Bile salts not only induce Vops, such as VopA and VopC, but also the needle-like secretion apparatus, including components such as T3SS2 translocon VopD2 (*Broberg et al., 2011*; *Zhang and Orth, 2013*; *Trosky et al., 2004*). In *V. parahaemolyticus*, the activation of T3SS2 by bile salts is regulated by two transmembrane ToxR-like transcription factors, VtrA (VPA1332) and VtrB (VPA1348) (*Gotoh et al., 2010*; *Kodama et al., 2010*). Homologues of VtrA and VtrB, named VttRA and VttRB respectively, have been identified in T3SS-containing non-O1/non-O139 *V. cholerae* strains and function in a similar way (*Alam et al., 2010*). Despite the identification of these transcription factors, the molecular

mechanism underlying bile salts sensing and T3SS2 signal propagation by *V. parahaemolyticus* and *V. cholerae* is unknown.

Here, we identify VtrC (VPA1333) from *V. parahaemolyticus* as a previously unrecognized component necessary for bile salt sensing and T3SS2 activation. *vtrC* is constitutively co-transcribed with *vtrA* as an overlapping 3′ open reading frame. Furthermore, we demonstrate that the periplasmic domains of the transmembrane proteins, VtrA and VtrC, form a functional complex that binds bile salts to activate VtrA's cytoplasmic DNA binding domain, which in turn induces T3SS2 via the downstream transcription factor VtrB. The structure of VtrA/VtrC periplasmic domains reveals an obligate heterodimer where VtrC recruits structural elements from VtrA to complete a β-barrel with a hydrophobic inner chamber that binds bile salts. A co-crystal structure of the VtrA/VtrC heterodimer with a bile salt reveals that ligand binding occurs in the hydrophobic inner chamber of VtrC, similar to that found for the family of monomeric calycins. Mutations of residues within the hydrophobic chamber that are important for bile salt-binding disrupt bile salt activation of T3SS2. Collectively, we uncover a mechanism by which bacteria can sense bile salts, and reveal how an evolutionarily conserved receptor senses an environmental cue to induce the production of virulence factors.

## Results

### V*trC* is conserved in various bacterial species with V*trA*-like genes and encodes a predicted transmembrane protein

*vtrC* (*vpa1333*) is a previously uncharacterized gene in *V. parahaemolyticus* that is located directly downstream of *vtrA* (*Figure 1A*). The open reading frames for *vtrA* and *vtrC* overlap by 17 nucleotides, suggesting that they are likely in the same operon and function in the same biological process. We observed that the gene organization of *vtrA*, *vtrC* and *vtrB* in the genome is highly conserved in other *Vibrio* and related species (*Figure 1A*). Both *V. cholerae* non-O1/O139 strains and *Grimontia hollisae* possess a T3SS similar to T3SS2 in *V. parahaemolyticus* and cause gastroenteritis during human infection (*Dziejman et al., 2005*). RT-PCR of the region spanning *vtrA* and *vtrC* showed that these genes are indeed in the same operon and co-transcribed before and after bile salt-mediated T3SS2 induction (*Figure 1B*, lanes 3, 5). PSI-BLAST analysis with the predicted VtrC protein sequence revealed that it is conserved in various *Vibrio* and related species, as well as in *Moritella*. VtrC has a highly conserved N-terminal transmembrane signal anchor that is predicted to direct and retain proteins in the bacterial membrane (*Figure 1C*). PSI-BLAST analysis using VtrA's sequence without the cytoplasmic DNA binding domain identified homologues in the same group of bacteria where VtrC is found (*Figure 1—figure supplement 1*). Furthermore, ten other strains of bacteria contain homologous genes of *vtrA* and *vtrC* but not *vtrB* (*Figure 1—figure supplement 2*). These results support the hypothesis that VtrA and VtrC evolved independently of VtrB.

### VtrC is essential for the activation of *V. parahaemolyticus* T3SS2 by bile salts

VtrA and VtrB are involved in the activation of the T3SS by bile salts. Therefore, we set out to determine whether VtrC also plays a role in this pathway. Because *vtrA* and *vtrC* overlap by 17 nucleotides, we generated a deletion of *vtrC* from the *V. parahaemolyticus* POR1 strain by retaining *vtrC*'s first 33 nucleotides thereby leaving the *vtrA* open reading frame intact (see Materials and methods). Deletion of *vtrC* completely abolished the activation of T3SS2 by bile salts, as shown by the loss of expression and secretion of the T3SS2 effectors VopA and VopC, and of the T3SS2 translocon VopD2 (*Figure 2A,B*). Complementation of *vtrC* deletion by a vector expressing VtrC fully restored the activity of T3SS2 (*Figure 2A,B*), indicating that the phenotypes observed in the *vtrC* deletion were not caused by a polar effect on neighboring genes. The effect of VtrC is specific to T3SS2 because its deletion had no impact on the expression and secretion of the T3SS1 effector VopS (*Figure 2C*). Taken together, these results indicate that VtrC is required for T3SS2 activation by bile salts.

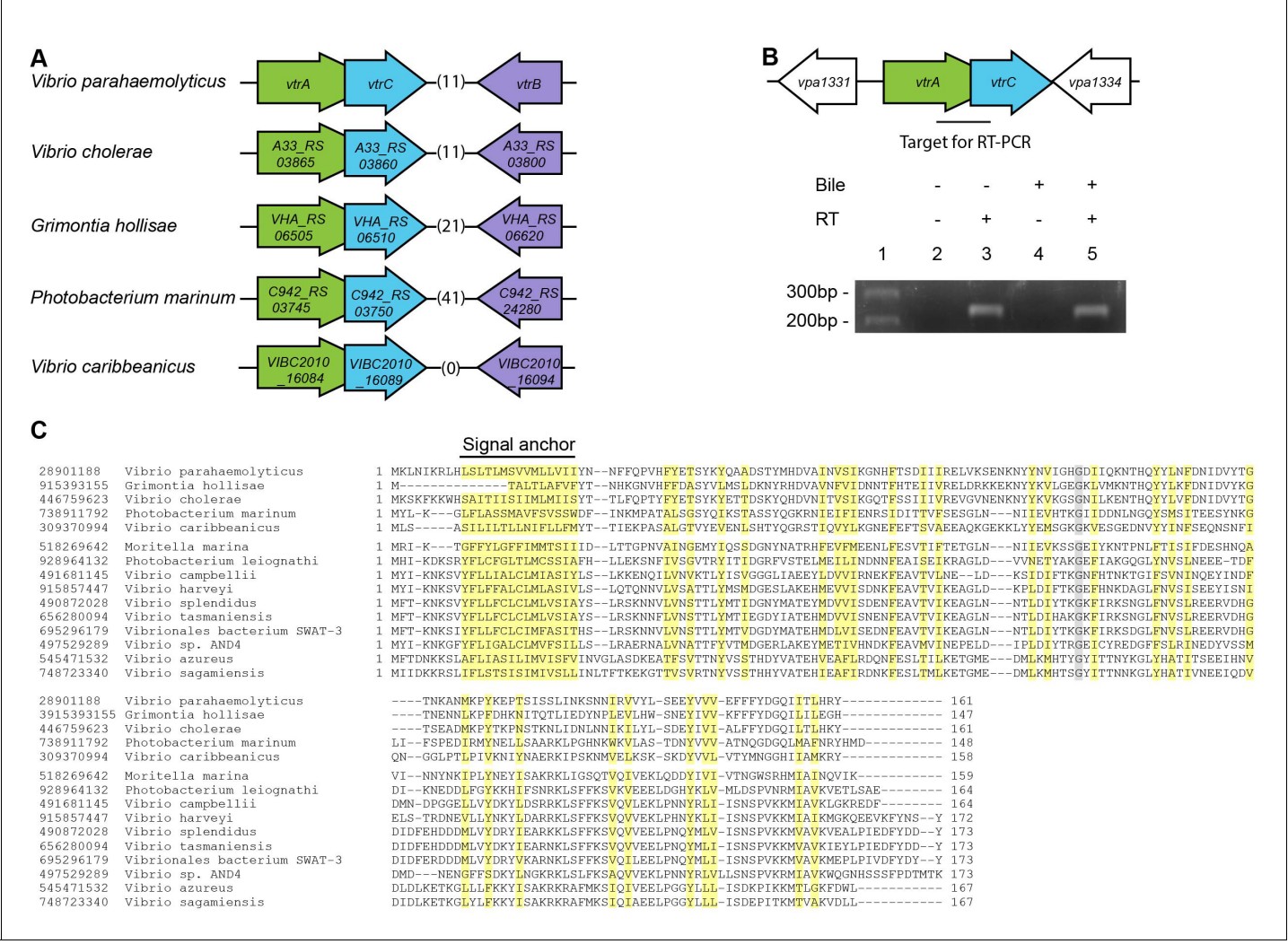

**Figure 1.** VtrC is conserved in various *Vibrio* and other species with VtrA-like sequences. (**A**) The gene organization of *vtrA*, *vtrC*, and *vtrB* is conserved in the T3SS2-like pathogenicity island of *Vibrionaceae* family species, with variable numbers of inserted genes (indicated by numbers in parentheses) between *vtrC* (or its homologous gene) and *vtrB* (or its homologous gene). (**B**) RT-PCR to amplify the target region shown by the bar that spans *vtrA* and *vtrC*. - Bile, *Vibrio parahaemolyticus* POR1 strain growing in LB without bile salts; + Bile, POR1 growing in LB supplemented with 0.05% bile salts. - RT, without reverse transcriptase; + RT, with reverse transcriptase; lane 1: DNA marker. RT-PCR is representative of three independent experiments. (**C**) Multiple protein sequence alignment of VtrC and its homologues. GI number of each protein is listed before the species names. Residues are highlighted according to group-wise conservations: mainly hydrophobic (yellow) and small (gray). Signal anchor: the predicted N-terminal transmembrane domain. An empty line is inserted between the species that contain (top) or lack (bottom) *vtrB* (or *vtrB* homologous gene). *marks protein sequences that were translated from nucleotide to include the entire N-terminus.

The following figure supplements are available for figure 1:

**Figure supplement 1.** Multiple protein sequence alignment of VtrA and its homologues.

**Figure supplement 2.** The gene organization of *vtrA* and *vtrC* in species that lack *vtrB*.

## VtrC localizes to the inner membrane with the N-terminal peptide in the cytoplasm and the C-terminal domain in the periplasm

Analysis of the VtrC protein sequence using membrane topology prediction programs, TMpred, TMHMM 2.0, HMMtop 2.0, and Phobius indicates VtrC is an inner membrane protein with the N-terminal 1–9 amino acids facing the cytoplasm, a single transmembrane helix (10–24 amino acids) and the C-terminal domain (25–161 amino acids) in the periplasm (*Figure 3A*). To verify the cellular

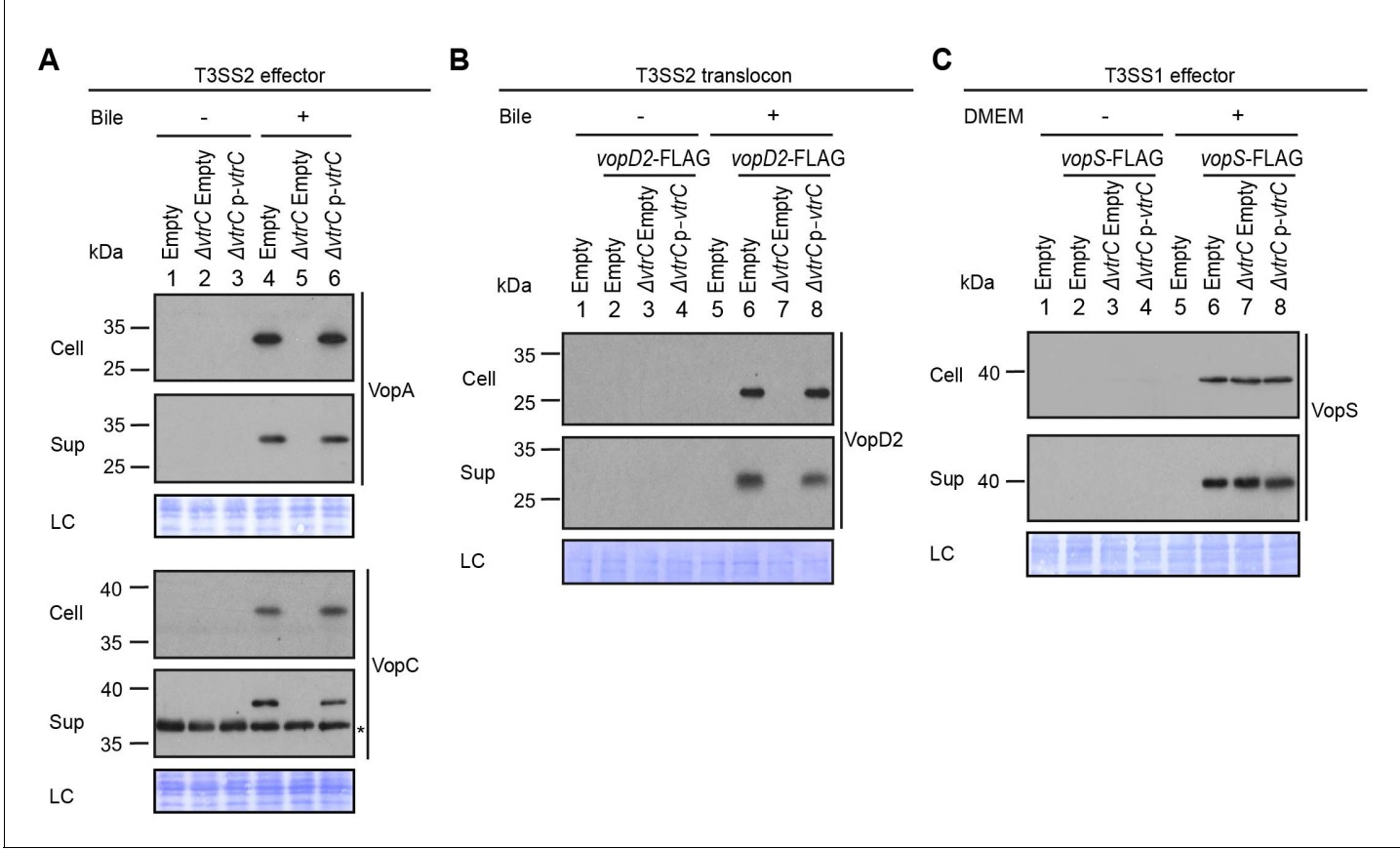

**Figure 2.** VtrC is essential for the activity of *V. parahaemolyticus* T3SS2 in the presence of bile salts. Expression (Cell) and secretion (Sup) of *V. parahaemolyticus* T3SS components were analyzed by Western blot. Loading control (LC) is shown for total protein lysate. (A–B) Expression and secretion of T3SS2 effectors VopA and VopC, and translocon VopD2 by *V. parahaemolyticus* POR1 derivative strains with the empty pBAD vector (Empty), *vtrC* deletion (Δ*vtrC* Empty) or *vtrC* complementation (Δ*vtrC* p-*vtrC*) containing a pBAD vector expressing VtrC under the putative promoter of its operon (1kb upstream of *vtrA*). Protein-specific antibodies were used to detect VopA and VopC. Anti-FLAG antibody was used to detect C-terminal endogenously FLAG-tagged VopD2. -/+ Bile, *V. parahaemolyticus* grown in LB without bile salts (-) or supplemented with 0.05% bile salts (+). Non-specific band is indicated with an asterisk. (C) Expression and secretion of T3SS1 effector VopS by *V. parahaemolyticus* POR1 derivative strains with the empty pBAD vector (Empty), *vtrC* deletion (Δ*vtrC* Empty) or *vtrC* complementation (Δ*vtrC* p-*vtrC*). Anti-FLAG antibody was used to detect endogenously C-terminal FLAG-tagged VopS. -/+ DMEM, *V. parahaemolyticus* grown in LB (-) or DMEM (+). Data is representative of three independent experiments.

localization and orientation of VtrC, we used two reporter proteins from *Escherichia coli*, alkaline phosphatase PhoA and β-galactosidase LacZ, whose activities depend on their subcellular localization. PhoA is active only in the periplasm after disulfide bond formation and dimerization, whereas LacZ only exhibits enzymatic activity in the cytoplasm where it can fold properly (*van Geest and Lolkema, 2000*). PhoA or LacZ were fused to either the N-terminus or the C-terminus of VtrC and their activities were then measured to determine the localization of VtrC (*Liu, 2003*). As expected, PhoA was active when fused to the C-terminus of VtrC, but not N-terminus (*Figure 3B*). In addition, LacZ was more active when fused to the N-terminus of VtrC (*Figure 3C*). These observations demonstrate that VtrC is localized to the inner membrane with the N-terminal 9 amino acids in the cytoplasm and the C-terminal domain in the periplasm.

## VtrC, like VtrA, functions upstream of VtrB

Previous studies have shown that VtrA directly controls the expression of VtrB and that both proteins are inner membrane proteins with their N-terminal DNA binding domain oriented in the cytoplasm and their C-terminal region in the periplasm (*Figure 4A*) (*Kodama et al., 2010*). Given that VtrC is constitutively expressed and its C-terminal domain is oriented in the periplasm, we hypothesized

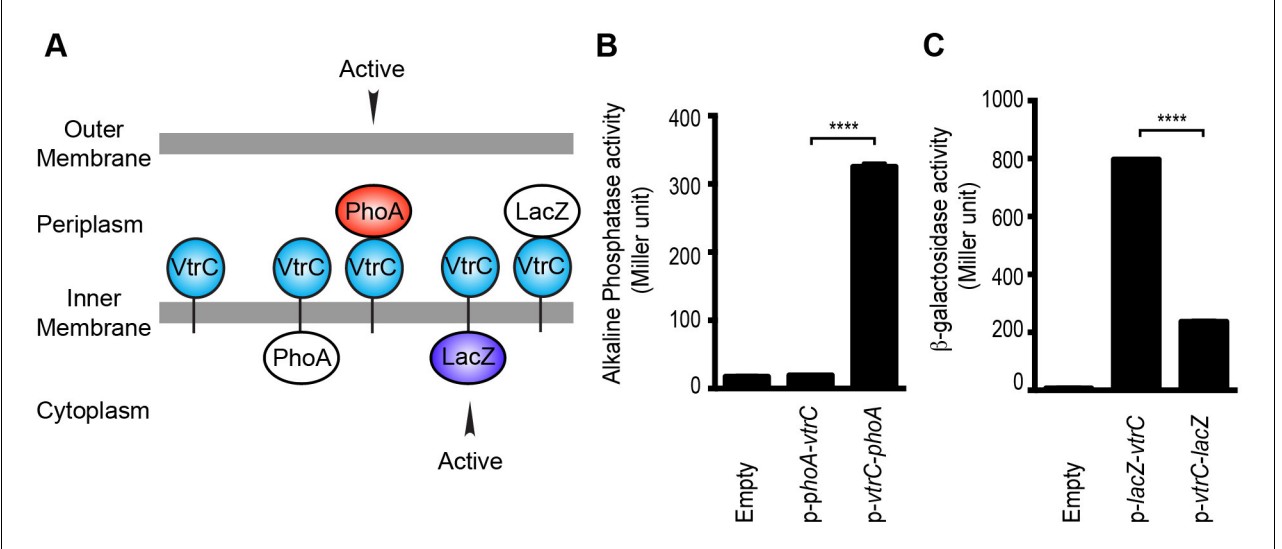

**Figure 3.** VtrC is an inner membrane protein with the N-terminus in the cytoplasm and the C-terminus in the periplasm. (**A**) Predicted cellular localization and orientation of VtrC. Active form of PhoA or LacZ fused to VtrC based on prediction. (**B**) Alkaline phosphatase PhoA activity of POR1 with the empty pBAD vector (Empty), expressing N terminal PhoA-VtrC (p-*phoA-vtrC*) or C terminal VtrC-PhoA (p-*vtrC-phoA*) fusion protein. (**C**) β-galactosidase LacZ activity of POR1 with the empty pBAD vector (Empty), expressing N terminal LacZ-VtrC (p-*lacZ-vtrC*) or C terminal VtrC-LacZ (p-*vtrC-lacZ*) fusion protein. ****p<0.0001, n = 3, +/- S.E.M. Data is representative of three independent experiments.

that VtrC, like VtrA, functions upstream of VtrB. The deletion of *vtrC* did not change the mRNA level of VtrA (*Figure 4B*). Interestingly, deletion of *vtrC* resulted in decreased protein levels of VtrA independent of bile salts (*Figure 4C*, lanes 3 and 7), suggesting that the presence of VtrC might stabilize VtrA. Ectopic expression of VtrC from a plasmid rescued VtrA levels (*Figure 4C*, lanes 4 and 8). As was previously observed for VtrA, the absence of VtrC prevented the induction of *vtrB* upon T3SS2 activation with bile salts (*Figure 4D*). Thus, VtrC appears to function upstream of VtrB via a signaling cascade including VtrA. The observed degradation of VtrA in the absence of VtrC suggested that there might be a direct physical interaction between VtrA and VtrC.

## VtrA and VtrC form a complex

To test whether VtrA and VtrC form a complex in vivo, we performed co-immunoprecipitation using endogenous VtrA and ectopically expressed N-terminal FLAG-tagged VtrC. Analysis of FLAG immunoprecipitates revealed that endogenous VtrA co-precipitated with VtrC, indicating that these two proteins interact in vivo (*Figure 5A*, lanes 3 and 6). VtrA and VtrC interact in the presence and absence of bile salts, suggesting that these proteins interact independently of bile salts before activation of T3SS2 by bile salts, and remain in a complex after treatment with bile salts (*Figure 5A*). Notably, ectopic expression of FLAG-VtrC, but not FLAG-VtrC that is deleted for its N-terminal transmembrane domain (FLAG-VtrCΔN30), fully complemented the deletion of *vtrC* and restored the activity of T3SS2, confirming that the wild type FLAG-tagged is functional (*Figure 5—figure supplement 1*). However, we noticed that the apparent molecular weight of full-length FLAG-VtrC (12 kDa) is smaller than the expected size of this protein (20 kDa) suggesting that VtrC may be processed in the periplasm. In any event, our results support that VtrC and VtrA interact in vivo.

Based on the orientation of VtrA and VtrC, we hypothesized these proteins would interact through their periplasmic domains. We found that the VtrA periplasmic domain (aa 161–253) is soluble when expressed in *E. coli*; however, attempts to express and purify the periplasmic domain of VtrC were unsuccessful. We predicted that the VtrA/VtrC interaction might stabilize the VtrC periplasmic domain. Therefore, we co-expressed the periplasmic domains of these two proteins in *E. coli*: amino acids 161–253 of VtrA and N-terminal His-tagged 31–161 of VtrC. Both proteins co-purified by Ni-NTA-affinity chromatography and remained as a stable and soluble complex during size exclusion chromatography (SEC). The VtrA/VtrC complex eluted as a 29.5 kDa species by SEC,

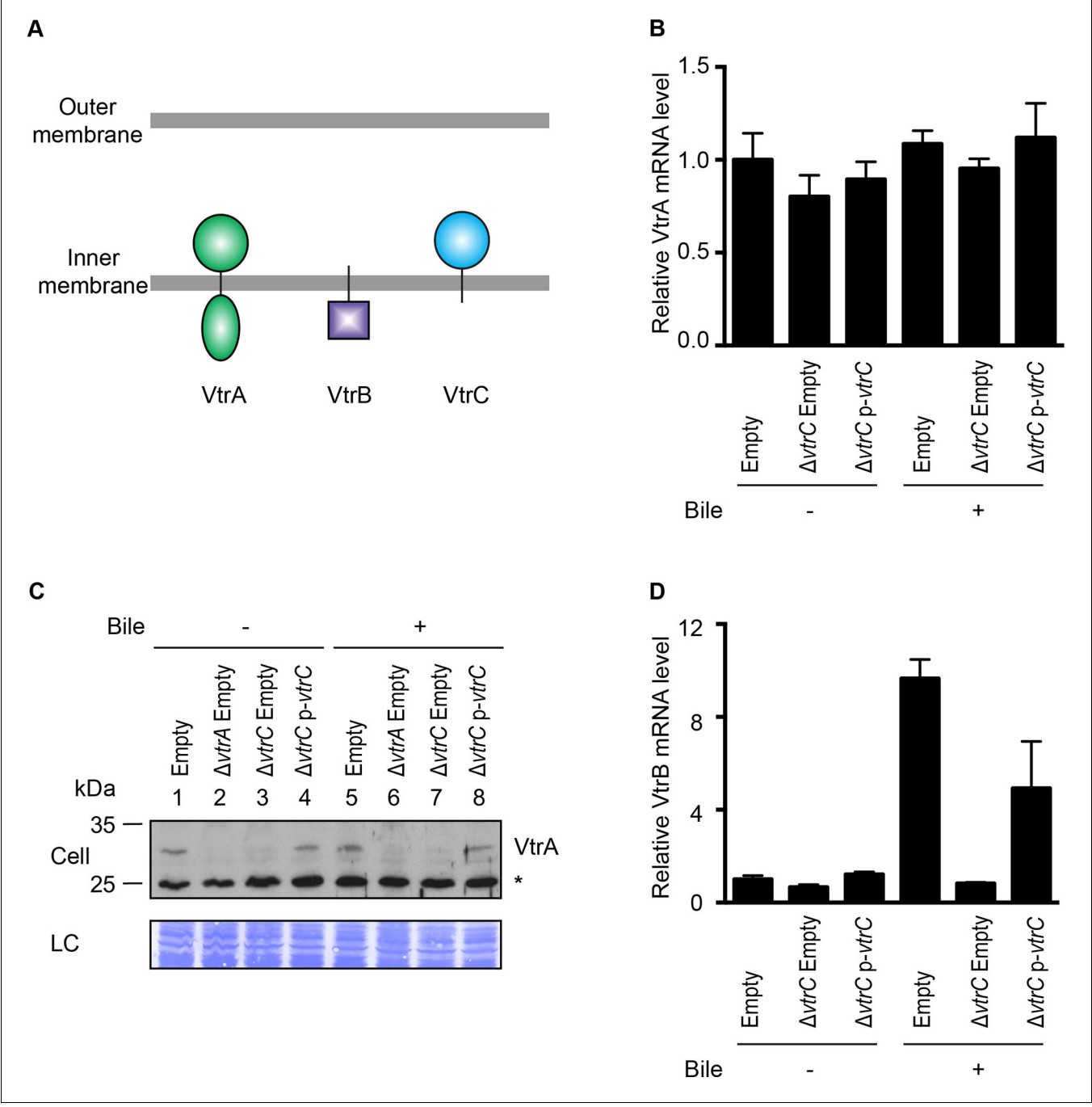

**Figure 4.** VtrC is necessary for maintaining VtrA protein level and the induction of VtrB. The effects of VtrC on VtrA and VtrB were characterized using *V. parahaemolyticus* POR1 derivative strains with the empty pBAD vector (Empty), *vtrA* deletion (ΔvtrA Empty), *vtrC* deletion (ΔvtrC Empty) or *vtrC* complementation (ΔvtrC p-vtrC) containing a pBAD vector expressing VtrC under the putative promoter of its operon (1kb upstream of *vtrA*). (**A**) Cellular localization and orientation of VtrA, VtrB and VtrC. (**B**) qRT-PCR analysis of VtrA mRNA level relative to POR1 with the empty pBAD vector (Empty) grown in LB without bile salts. (**C**) Western blot analysis of VtrA protein level. Protein specific antibody was used to detect VtrA. Non-specific band is indicated with an asterisk. (**D**) qRT-PCR analysis of VtrB mRNA level relative to POR1 with the empty pBAD vector (Empty) grown in LB without bile salts. -/+ Bile, *V. parahaemolyticus* grown in LB without bile salts (-) or supplemented with 0.05% bile salts (+). For qRT-PCR analysis, expression of *vtrA* and *vtrB* was normalized to the expression of the control gene *fliA*. Data is representative of three independent experiments.

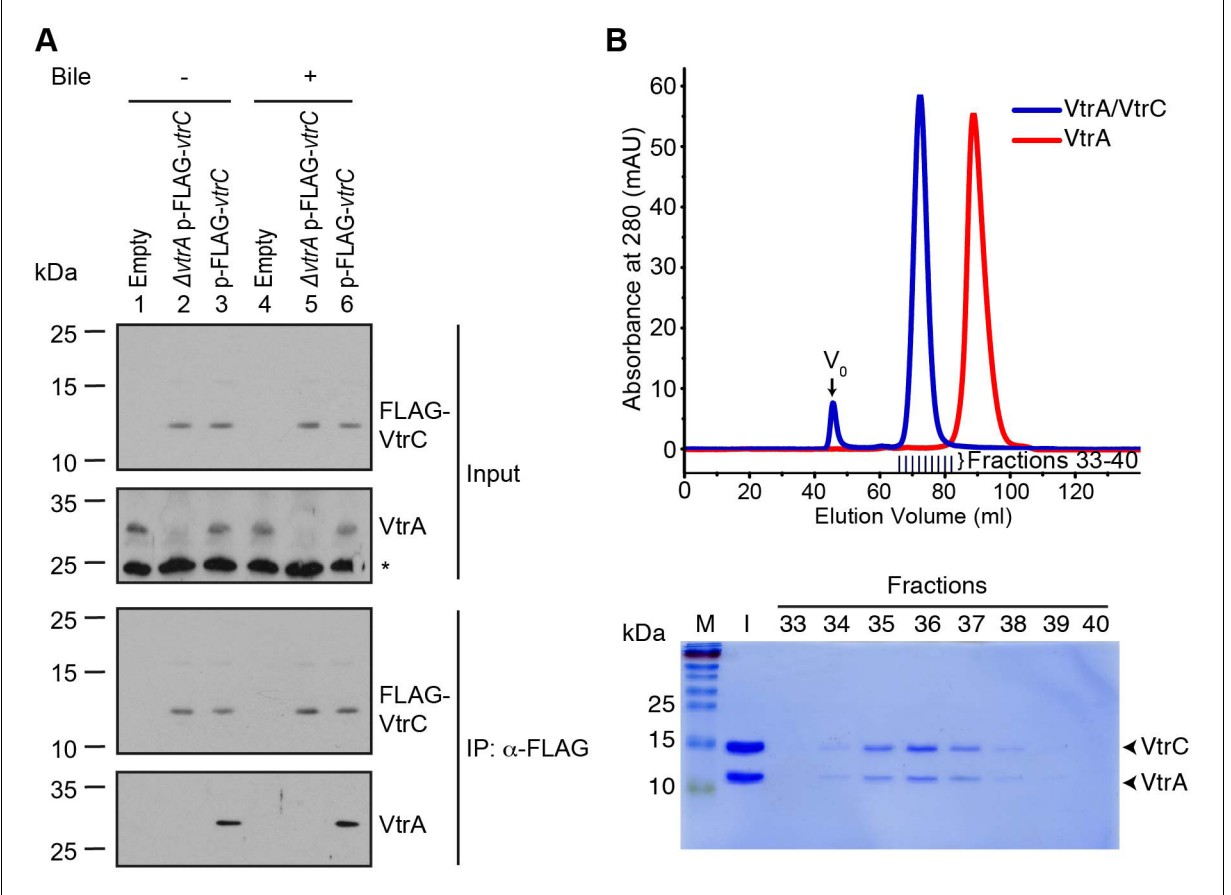

**Figure 5.** VtrA and VtrC form a complex. (**A**) Co-immunoprecipitation (Co-IP) of endogenous VtrA and vector induced FLAG-VtrC. pBAD vector induced N-terminal FLAG-tagged VtrC was immunoprecipitated with anti-FLAG affinity gel from *V. parahaemolyticus* POR1 derivative strains that express only VtrA (Empty), only N-terminal FLAG-tagged VtrC (Δ*vtrA* p-FLAG-*vtrC*), or both (p-FLAG-*vtrC*). Protein-specific antibody was used to detect VtrA. Anti-FLAG antibody was used to detect FLAG-VtrC. -/+ Bile, *V. parahaemolyticus* grown in LB without bile salts (-) or supplemented with 0.05% bile salts (+). Non-specific band is indicated with an asterisk. Data is representative of three independent experiments. (**B**) Top, gel filtration analysis of the VtrA/VtrC periplasmic domain complex, void volume of the column is indicated as $V_0$. Bottom, SDS-PAGE analysis of samples from the fractions corresponding to the elution peak of the complex.

The following figure supplement is available for figure 5:

**Figure supplement 1.** N-terminal FLAG-tagged VtrC is functional.

indicating a 1:1 heterodimer of VtrA (11.0 kDa) and VtrC (16.9 kDa) periplasmic domains (*Figure 5B*). VtrA alone appeared to elute as a monomer of approximately 11 kDa (*Figure 5B*).

## The VtrA/VtrC complex is an obligate heterodimer

To further understand the nature of the interaction between the periplasmic domains of VtrA and VtrC, we crystallized the complex and obtained its X-ray structure. The structure was solved by multiple-wavelength anomalous dispersion phasing using anomalous signals from selenomethionine and refined to a resolution of 2.70 Å using native data from an isomorphous crystal. Two of the five selenomethionines were used for phasing. The two N-terminal methionines and Se-Met 115 in the loop that covers the beta barrel of VtrC were not observed in the final electron density map and were not included in the final model. The asymmetric unit contains one complex with one copy each of the VtrC and VtrA periplasmic domains (*Figure 6A*). The VtrA and VtrC subunits in the heterodimer make extensive interactions with each other (*Figure 6B*), with an interface area of 1,149 Å$^2$

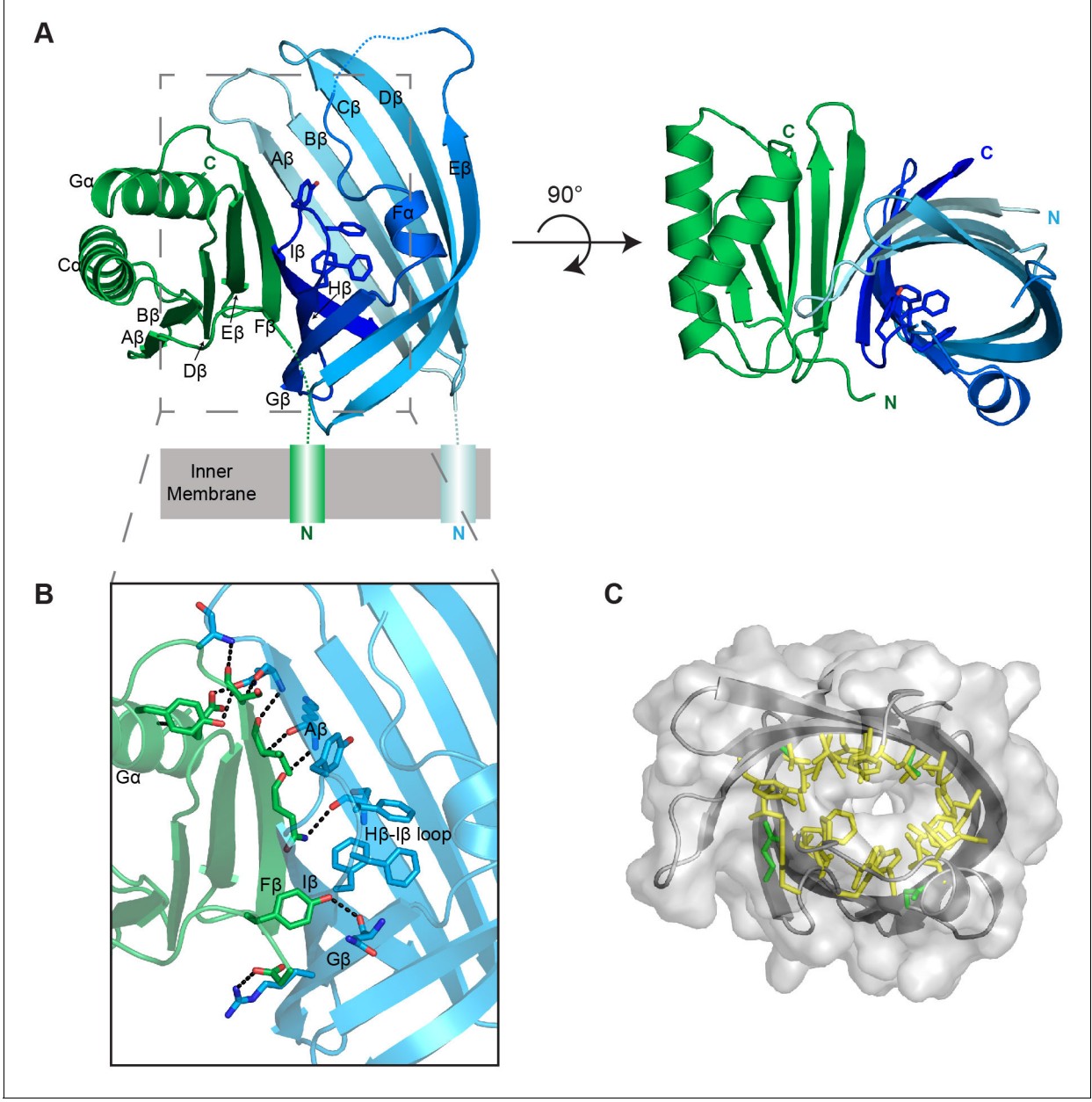

**Figure 6.** Structure of the VtrA/VtrC heterodimer. (**A**) Cartoon representation of the periplasmic domain complex formed by VtrA (green) and VtrC (blue, light to dark gradient from N-terminus to C-terminus). Side chains of Hβ-Iβ loop residues are shown as sticks. (**B**) Detailed view of the VtrA/VtrC interface. Selected residues that form polar contacts (black dashed lines), as well as potential bile salt binding residues are shown as sticks. (**C**) Overlay of surface and ribbon models of VtrC showing interior cavity. Side chains of residues lining the cavity are shown as sticks in yellow for hydrophobic residues (Ala, Val, Ile, Leu, Met, Phe, Tyr, Trp) and green for all other.

The following figure supplement is available for figure 6:

**Figure supplement 1.** Activation of T3SS2 by individual bile acid.

---

(***Krissinel and Henrick, 2007***). An analysis of the macromolecular interfaces in this crystalline lattice by the web server PDBePISA (http://www.ebi.ac.uk/pdbe/pisa/pistart.html) (***Krissinel and Henrick, 2007***) indicate that the VtrA/VtrC heterodimer is the only stable quaternary structure in solution.

The VtrC subunit consists of eight β-strands and a short α-helix. The eight β-strands of VtrC form a β-sheet meander that folds into itself to form a β-barrel with a small space between the first and

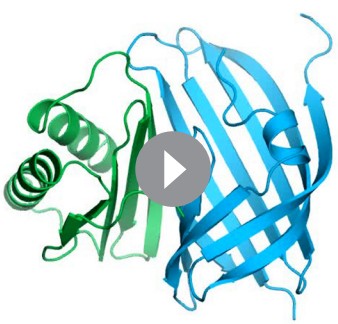

**Video 1.** Structure of the VtrA/VtrC heterodimer. Rotation of VtrA/VtrC heterodimer demonstrates the binding cavity of the beta-barrel. Further rotation shows how VtrA contributes a strand to the 'open' beta barrel formed by VtrC. Extensive hydrogen bonding is observed between VtrA and VtrC.

last β-strands (Aβ and Iβ) (*Video 1*). The β-barrel is lined on the inside by hydrophobic side chains (*Figure 6C*) and is covered on one side by the short α-helix (Fα) and an adjacent disordered loop, which could not be modeled in its entirety due to the poor electron density of this region (*Figure 6A*). The Hβ-Iβ loop is particularly rich in aromatic side-chains, promoting us to hypothesize that this loop could be involved in binding the steroid ring structure of bile acids (*Figure 6A,B*). Despite the lack of sequence similarity to proteins of known function, VtrC shares structural similarity with lipocalins, fatty acid binding proteins (FABPs), avidins, metalloprotease inhibitors (MPIs), and other members of the calycin superfamily of proteins (*Figure 7*) (*Flower et al., 2000*). The inside of the calycin β-barrel often acts as a binding site for fatty acids and other hydrophobic molecules, such as retinol and biotin (*Flower et al., 2000*). Comparison of the overall VtrC structure with other calycins suggests that it adopts a new fold within the superfamily. VtrC falls in between the lipocalins, which include a C-terminal helix that packs against the barrel on the corresponding VtrA interacting surface; and the FABPs, which include a β-hairpin insertion into the typical 8-stranded β-barrel in the position of the VtrC disordered loop. Interestingly, VtrA/VtrC would be the first member of this family that is an obligate dimer and not only binds a hydrophobic ligand but also transmits a signal upon binding.

The VtrA subunit adopts an alpha/beta fold with a five-stranded β-sheet and two α-helices on one side of the sheet (*Figure 6A*). The bulk of the contacts with VtrC is through VtrA's five-stranded β-sheet on the side opposite to the α-helices and involves both hydrophobic and polar contacts. VtrA Fβ intercalates between VtrC Aβ and Iβ, making numerous polar contacts with VtrC Aβ and closing the gap in the VtrC β-barrel (*Figure 6B*). The incorporation of structural elements from VtrA into the β-barrel fold suggests that VtrC may not fold properly in isolation and is consistent with our inability to purify the VtrC periplasmic domain in the absence of VtrA. Interestingly, the N-terminus of the VtrA periplasmic domain, leading to the transmembrane helix, contacts VtrC Gβ and is in close proximity to the aromatic-rich Hβ-Iβ loop (*Figure 6B*), leading us to hypothesize that these residues might be involved in signal transmission across the membrane in response to ligand binding.

## The VtrA/VtrC complex binds bile salt

Based on the structural similarities between the monomeric calycins and the VtrA/VtrC heterodimer, we predicted that VtrA/VtrC complex could bind bile salts. Previously, Gotoh et al. (*Gotoh et al., 2010*) established that bile salts are the component of bile that activates the T3SS2. We have recapitulated this data and confirmed which purified bile salts could activate T3SS2. We validated that taurodeoxycholate (TDC) and glycodeoxycholate (GDC), but not chenodeoxycholate (CDC) or cholate (CA), could activate the *V. parahaemolyticus* T3SS2 (*Figure 6—figure supplement 1*).

Based on this information we moved forward with biophysical experiments to test whether the VtrA/VtrC heterodimer is the bile salt receptor using a relevant bile salt, TDC. We first tested whether the VtrA/VtrC periplasmic domain heterodimer could bind bile salts using isothermal titration calorimetry (ITC). Negative power deflections were observed throughout the titration of the bile salt TDC into the VtrA/VtrC solution. These results indicated that TDC binds to the VtrA/VtrC heterodimer in an exothermic manner (*Figure 8*), with a dissociation constant ($K_D$) of 315.4 nM (*Figure 8A*). The stoichiometry of TDC binding to VtrA/VtrC is approximately 1:1 (n = 0.94). Taken together our results suggest that VtrA and VtrC form a functional complex that can bind bile salts.

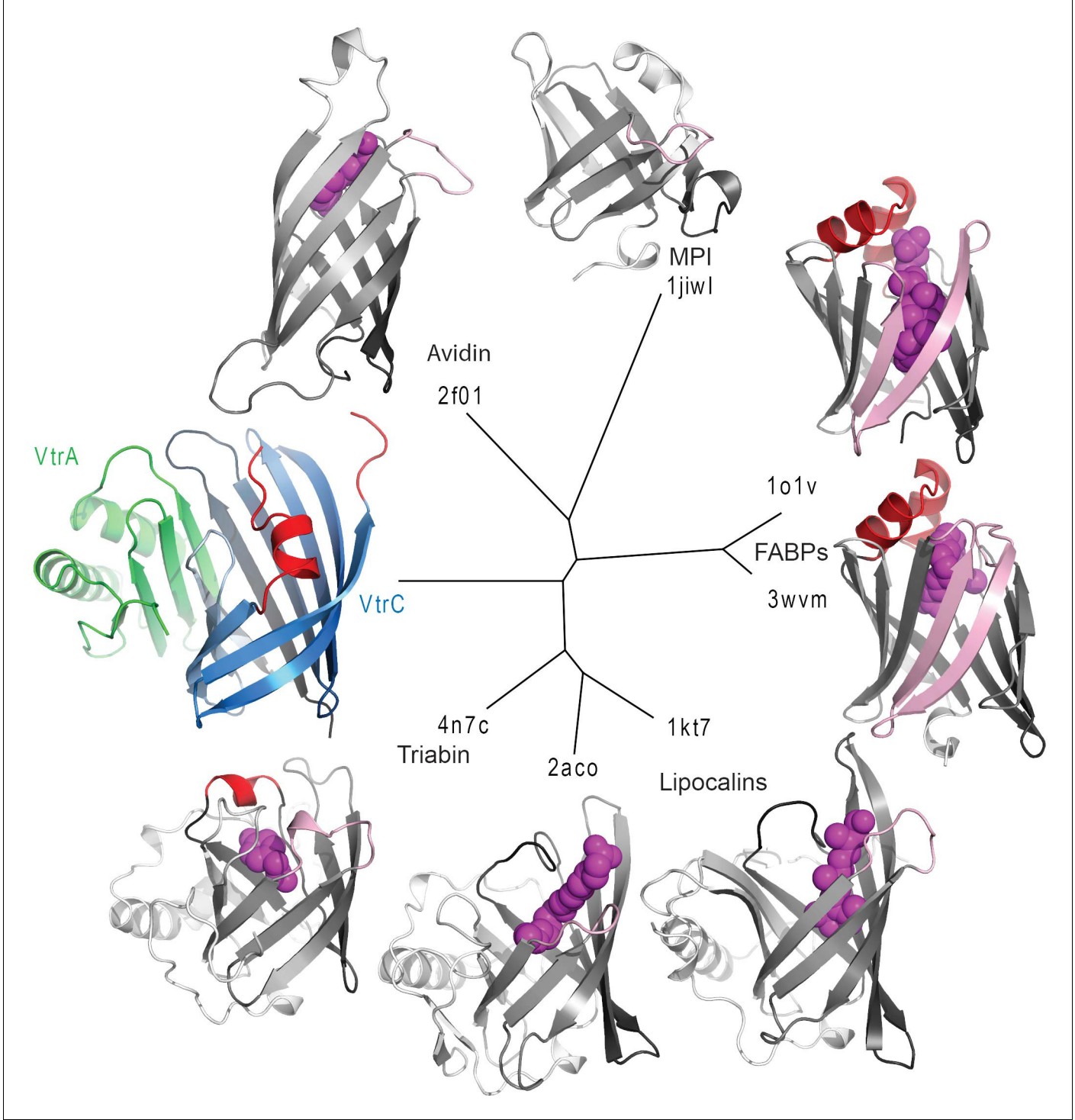

**Figure 7.** Structure based distance tree of VtrA/VtrC heterodimer and members of the calycin superfamily. Structures representatives of calycin superfamilies: MPI, avidin, FABPs, lipocalins, and triabin with bound ligands in the centers of the barrels (magenta spheres) were chosen. The conserved 8-stranded barrel core found in all structures is colored from dark to light grey for all representatives and dark to light blue for VtrC. The Hβ-Iβ loop containing the presumed VtrC helical lid (red) is unique to the subunit, as compared to the corresponding loops from representative structures (light pink). This loop also corresponds to the position of the inserted FABP β-hairpin (light pink). The functionally analogous helical lid in FABPs (red) is located after the first β-strand of the barrel. The triabin structure closes the binding pocket with hydrophobic residues from both the FABP lid loop (red) and the VtrC loop (pink).

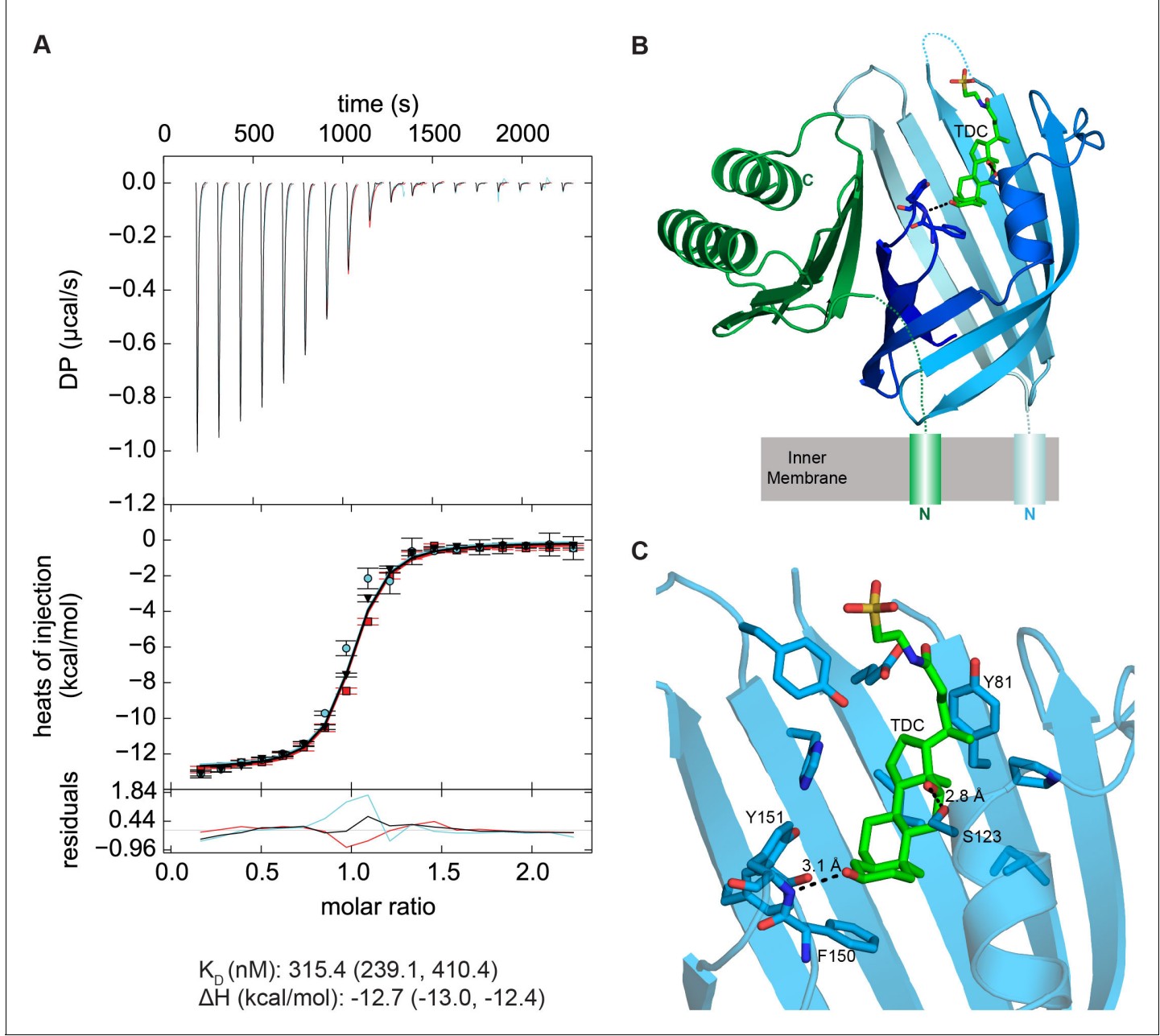

**Figure 8.** VtrA/VtrC periplasmic domain complex binds the bile salt taurodeoxycholate (TDC). (**A**) ITC-derived binding curves of the VtrA/VtrC complex with TDC. Thermodynamic parameters were determined by global fitting of triplicate isotherms (presented in black, red, and cyan). The dissociation constant ($K_D$) and enthalpy ($\Delta H$) values are reported followed by the $1\sigma$ error intervals in parenthesis. Data is representative of two independent experiments. (**B**) Structure of the VtrA/VtrC periplasmic domain complex binding TDC (green). VtrA and VtrC follow same coloring scheme as in *Figure 6*. (**C**) Detailed view of the TDC binding site. Hydrogen bonds between protein and TDC are represented as dashed lines.

The following figure supplements are available for figure 8:

**Figure supplement 1.** Electron density around TDC molecules.

**Figure supplement 2.** Comparison between apo and TDC bound structure heterodimers.

## VtrA/VtrC binds bile salts in a primarily hydrophobic cleft in the VtrC $\beta$-barrel

We next purified the VtrA/VtrC heterodimer in the presence of the bile salt TDC and crystalized this complex and obtained its X-ray structure. The crystal structure was solved via molecular replacement by using our original VtrA/VtrC heterodimer as a search model, and refined to a resolution of 2.10 Å. The model reveals three VtrA/VtrC heterodimers in the asymmetric unit, each with clear electron density for one TDC molecule bound inside the β-barrel (*Figure 8B*, *Figure 8—figure supplement 1A*). A fourth TDC molecule mediated lattice contacts between VtrA chain A and its symmetry mate (*Figure 8—figure supplement 1B*). An analysis of the macromolecular interfaces found in this crystalline lattice by the web server PDBePISA (*Krissinel and Henrick, 2007*) indicated that the heterodimer is the only stable quaternary structure in solution. Although the three heterodimers in the asymmetric unit are highly similar, average root mean squared deviations (RMSD) of 0.5 Å for VtrA and 0.6–1.5 Å for VtrC indicates that there are differences in the VtrC monomer (*Figure 8—figure supplement 2A, E*). RSMD values of 0.4–0.5 Å for VtrA and 1.2–1.9 Å were obtained by superimposing each of the TDC-bound heterodimers to the apo structure (*Figure 8—figure supplement 2B--E*), indicating that the overall fold is maintained but there are considerable changes in the VtrC monomer. The largest differences between the VtrA/VtrC-TDC and the apo heterodimers are in the VtrC disordered loop and Fα that follows it. In the VtrA/VtrC-TDC heterodimer, the disordered loop has been displaced by the TDC molecule and is now observable in two out of the three heterodimers in the crystallographic asymmetric unit. Residues Thr122 and Ser123, which were in the disordered loop in the apo structure, now form part of Fα in two of the TDC-bound heterodimers. Remarkably, one of these residues (Ser123) now forms a hydrogen bond with the 12α-hydroxyl group of TDC (*Figure 8C*). The main chain amide between F150 and Tyr151 (Hβ-Iβ loop) coordinates the 3α-hydroxyl group (*Figure 8C*), which is present in all bile salts, suggesting that this interaction is important for specificity towards bile salts. Several VtrC residues in the binding pocket maintain hydrophobic interactions with TDC, including the side chains of Phe150 with steroid ring A and Tyr81 with the hydrophobic part of the bile acid (*Figure 8C*).

## Mutations of the VtrA/VtrC complex with the bile salt TDC

Based on our co-crystal structure of VtrA/VtrC with TDC, the role of several residues that contact the bile salt were investigated for their ability to affect the binding of bile salts and activation of the T3SS2 (*Figure 9*). The mutation of His50, located at the top of the hydrophobic cleft, to an arginine residue is expected to produce steric hindrance by insertion of a large side chain into the bile salt binding pocket (*Figure 9A*). When VtrC-H50R was expressed in the *vtrC* deletion strain and bile salts were added, the T3SS2 could not be induced as indicated by the lack of expression and secretion of the T3SS2 effector VopA (*Figure 9B*, lanes 5 and 11). We also changed Tyr81 to an alanine residue, which we predicted would disrupt bile salt binding by eliminating the interaction between the Tyr81 aromatic ring and the bile salt side chain (*Figure 9A*). When VtrC-Y81A was expressed in the *vtrC* deletion strain and bile salts were added, the T3SS2 was not induced as indicated by the lack of expression and secretion of VopA (*Figure 9B*, lanes 6 and 12). As a control, we chose to mutate a surface exposed residue, Gln42, to an alanine (*Figure 9A*). As expected, this mutant rescued the *vtrC* deletion strain and upon addition of bile salts the T3SS2 was activated as indicated by the expression and secretion of VopA (*Figure 9B*, lanes 4 and 10). To assess whether these mutations had any effect on the VtrA/VtrC complex formation, we co-immunoprecited the complex with the N-terminal FLAG-tagged VtrC. As seen in *Figure 9C*, the wild type and mutant VtrC proteins formed a stable complex with endogenous VtrA, as indicated by their co-immunopreciptation.

## Discussion

Bile sensing plays a significant role in human infection caused by various enteric bacterial pathogens (*Begley et al., 2005*). Deciphering how bacteria use bile as a signal to regulate the expression of virulence genes is essential to our understanding of disease mechanisms and also can offer insights for designing novel therapeutic strategies. Bile salt-induced T3SSs are the primary virulence factor in many *Vibrio* species (*Broberg et al., 2011*). In this study, we uncovered a previously unidentified component of a widespread, conserved signaling platform that is essential for bile salt sensing and

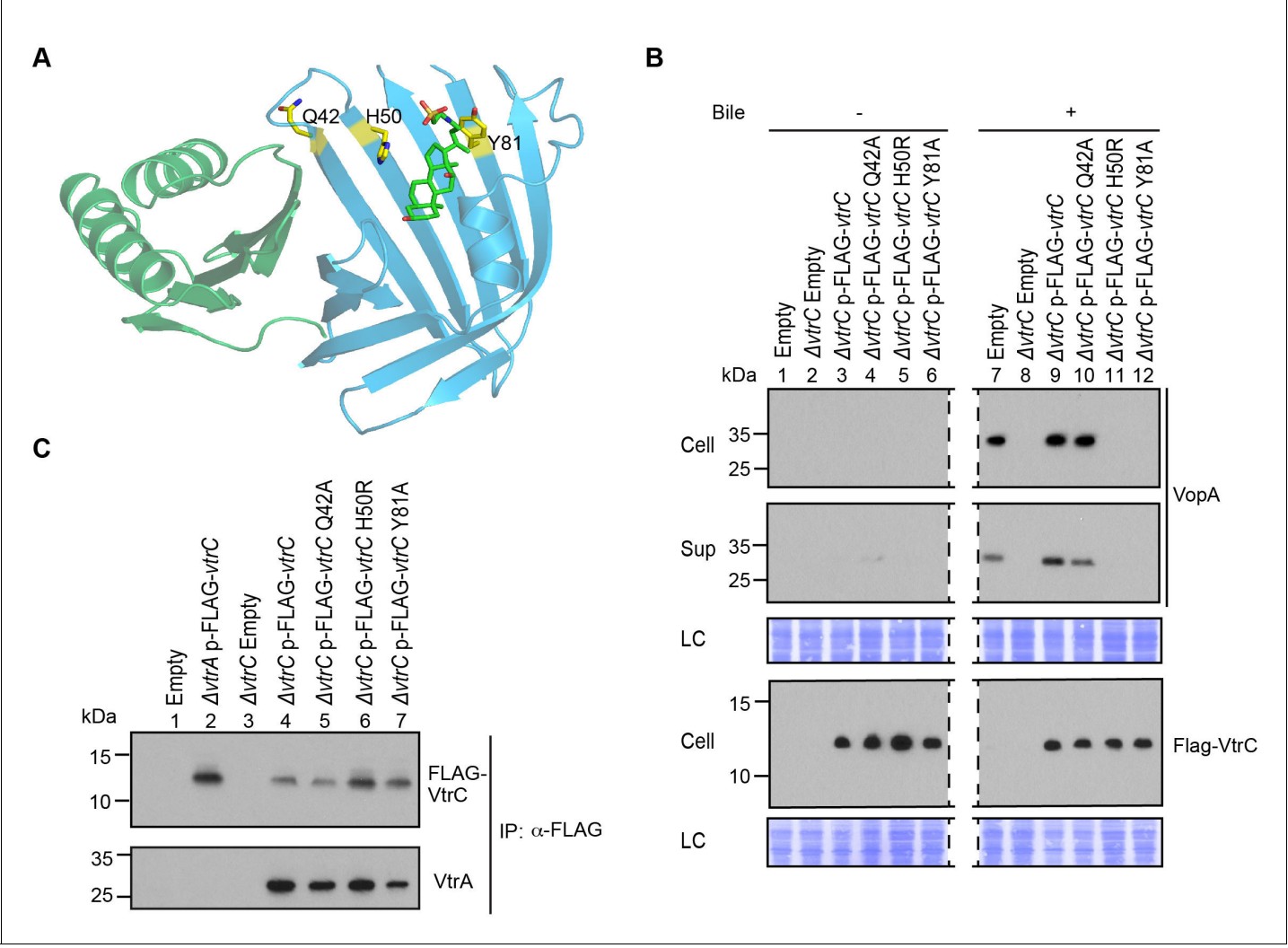

**Figure 9.** Mutations in the hydrophobic chamber of VtrA/VtrC heterodimer disrupt signaling mediated by bile salts. (A) Structure of VtrA/VtrC heterodimer. Q42, H50 and Y81 of VtrC are highlighted in yellow. (B) Restoration of T3SS2 activation in POR1ΔvtrC by VtrC mutant Q42A but not H50R or Y81A. Expression (Cell) and secretion (Sup) of *V. parahaemolyticus* T3SS2 effector VopA by POR1 derivative strains with the empty pBAD vector (Empty), *vtrC* deletion (ΔvtrC Empty), *vtrC* complementation by N-terminal FLAG-tagged wild type (ΔvtrC p-FLAG-*vtrC*) or mutant (Q42A, H50R or Y81A) VtrC that expresses pBAD vector induced protein under the arabinose inducible promoter. Protein-specific antibody was used to detect VopA. Anti-FLAG antibody was used to detect N-terminal FLAG-tagged VtrC. -/+ Bile, *V. parahaemolyticus* grown in LB without bile salts (-) or supplemented with 0.05% bile salts (+). Loading control (LC) is shown for total protein lysate. Dashed lines indicate lane that was deleted from the gel. (C) Co-IP demonstrating that all three VtrC mutants form a stable complex with VtrA.

activation of virulence factors in *Vibrios*. Our biochemical data and the structural analysis demonstrated that the periplasmic domains of the inner membrane proteins VtrA and VtrC form a stable complex that functions as a receptor for bile salts.

The crystal structure of the VtrA/VtrC complex reveals that their periplasmic domains form a obligate heterodimer where VtrC folds into an open eight-stranded β-barrel and VtrA adopts a mixed α/β structure whereby one of its a β-strands contributes to the formation of the β-barrel. The VtrA/VtrC heterodimer fold is similar to members of the calycin superfamily, which contain eight to ten-stranded β-barrels that often bind hydrophobic molecules (*Flower et al., 2000*). Calycins have low sequence similarity, but relatively high structural conservation. They can be subdivided into several families based on structural details characteristic of each group (*Figure 7*) (*Flower et al., 2000*). Calycin families are markedly different in their distribution among biological kingdoms, cellular localization, and function, e.g.: (1) FABPs are cytosolic proteins involved in lipid homeostasis in animals,

(2) MPIs are bacterial proteins that are secreted into the periplasm to inhibit metalloproteases, (3) lipocalins are found in both eukaryotes and prokaryotes, and they are mostly extracellular or attached to the outer membrane and bind a variety of ligands (*Flower et al., 2000*). In terms of structure, VtrC does not seem to cluster with any particular family and it forms its own clade (*Figure 7*). Although calycins have been found to form complexes with other proteins, our inspection of the available structures of complexes did not reveal any arrangement that resembled that of the VtrA/VtrC obligate heterodimer. This complex is unique for this family because it is formed from two proteins and it is a complex that transmits a signal upon binding its ligand.

Interestingly, members of the FABP family have been found to bind bile salts. This is exemplified by human and porcine ileal lipid binding proteins (ILBPs), and their NMR structures in complex with taurocholate (*Kurz et al., 2003*) and glycocholate (*Lücke et al., 2000*), respectively. Analogous to VtrC's disordered loop and short α-helix, FABPs have a helix-turn-helix motif that forms a lid over one side of the barrel and is proposed to act as a portal for ligand access (*Storch and McDermott, 2009*). In the case of FABPs, it has been suggested that ligand binding induces subtle conformational changes that promote protein-membrane or protein-protein interactions (*Storch and McDermott, 2009*). We observe a similar scenario for the VtrA/VtrC obligate heterodimer bound to bile salts where, upon binding bile salt, it appears an unstructured loop is translocated from the inside of the hydrophobic chamber to the outside of the chamber, where it becomes structured. Future studies will focus on how a conformational change during bile salt binding may induce a switch in the VtrA/VtrC interaction that will result in activation of the VtrA cytoplasmic transcription activator domain (*Figure 9*).

The closest example with regard to complex formation and function is the GrlR/GrlA complex, a regulator of the T3SS-encoding locus of enterocyte effacement (LEE) in enterohaemorrhagic and enteropathogenic *E. coli* strains (EHEC and EPEC, respectively) (*Padavannil et al., 2013*). GrlR is a cytosolic lipocalin that represses the LEE repressor GrlA by binding to its helix-turn-helix (HTH) domain. Similar to *vtrA* and *vtrC*, *grlR* and *grlA* are transcribed from a bicistronic operon, although the order of the lipocalin and transcriptional regulator genes is reversed in the latter pair. Analytical ultracentrifugation and the structure of GrlR in complex with the GrlA HTH domain revealed a heterotrimeric complex where a GrlR dimer binds one GrlA HTH monomer (PDB ID: 4KT5) (*Padavannil et al., 2013*). It has been shown that the inner cavity of GrlR binds lipids (*Jobichen et al., 2009*). However, the role of lipid binding in this system is unknown. Although GrlR and VtrC have analogous roles regulating T3SS through their interaction with a transcription factor, they do so from different cellular compartments at different stoichiometries, and their binding partners (GrlA HTH domain and VtrA periplasmic domain, respectively) have different folds.

Several signaling features of VtrA/VtrC appear to parallel two other pairs of regulatory proteins found in *V. cholerae*, ToxR/ToxS and TcpP/TcpH, which control the expression of numerous virulence factors including CT and TCP. First, the genes encoding the proteins in each pair are arranged in bicistronic operons (*Miller et al., 1989*; *Häse and Mekalanos, 1998*). Second, VtrA/ToxR/TcpP and VtrC/ToxS/TcpH have the same cellular localization and membrane topology (*Miller et al., 1989*; *Häse and Mekalanos, 1998*; *Miller et al., 1987*). Third, VtrC, ToxS and TcpH affect the activity of VtrA, ToxR and TcpP, respectively. On this last point, we showed that VtrA and VtrC form a complex in vivo both before and after T3SS2 activation and in vitro through their periplasmic domains.

Our data supports a model in which VtrA and VtrC form a complex both in the absence and presence of bile salts. Upon binding bile salts to the hydrophobic chamber in the VtrA/VtrC complex, the cytoplasmic DNA binding domain of VtrA is activated and in turn induces VtrB to activate the T3SS2 (*Figure 10*). The specific changes caused by binding bile salts and their effects on the VtrA DNA-binding domain are yet to be determined. Although VtrA contains no cysteine residues in its periplasmic domain, it is still possible that bile salts increase VtrA activity by inducing its dimerization or oligomerization, similar to what has been proposed for the ToxR/ToxS and TcpP/TcpH systems (*Ottemann and Mekalanos, 1996*; *Yang et al., 2013*).

The VtrA/VtrC complex is highly conserved in a group of diverse *Vibrionaceae* family species, as well as in *Moritella marina*. While the bile salt-induced T3SS in *V. cholerae* non-O1/O139 strains is necessary for bacterial colonization and the primary cause of disease during infection (*Chaand et al., 2015*), the role of T3SS in *G. hollisae* infection is yet to be characterized. For other bacteria encoding *vtrA*, *vtrC* and *vtrB*, such as *Photobacterium marinum* and *V. caribbeanicus,* only a few T3SS related genes or no T3SS related genes are observed in their genomes, respectively. The differences between

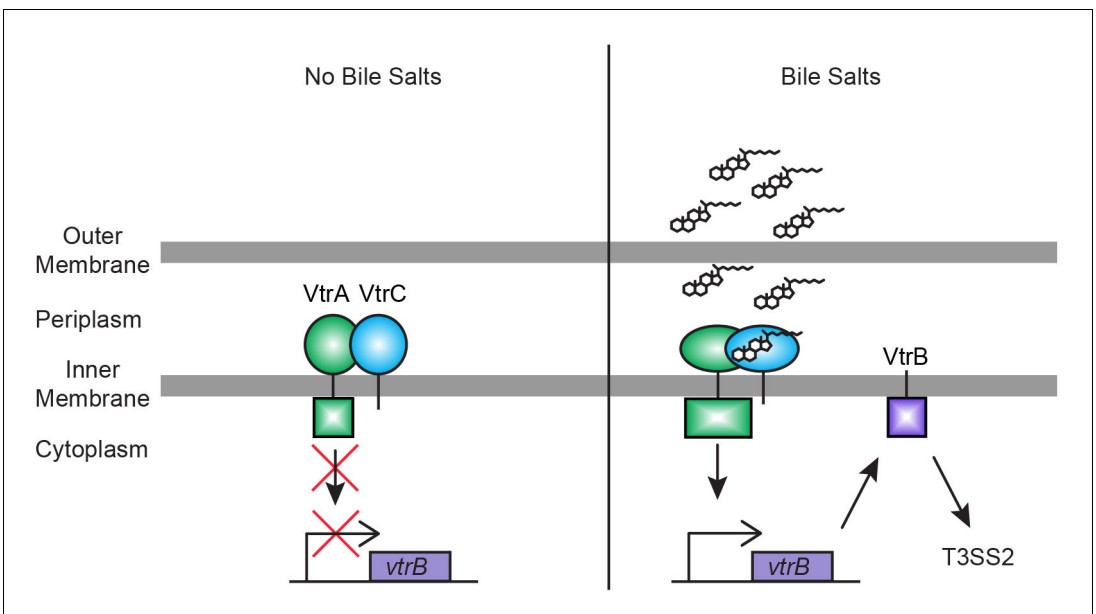

**Figure 10.** Model for bile salts sensing and T3SS2 regulation in *V. parahaemolyticus*. In the absence of bile salts, VtrA (green) and VtrC (blue) form a complex where the VtrA cytoplasmic DNA binding domain is kept inactive. Upon binding of bile salts, the VtrA/VtrC complex activates the VtrA DNA binding domain by a yet to be determined mechanism, which induces VtrB (purple) expression. VtrB, in turn, activates T3SS2.

T3SS gene representation in these bacteria demonstrate this system's diversity and imply that the T3SS may be evolving in response to different living environments. Based on surrounding genes, this bile salt receptor complex may have adapted in other species to bind other hydrophilic ligands. For example, in the ten strains that lack a *vtrB* homolog but possess a *vtrA/vtrC* pair, the gene downstream of the *vtrA/vtrC* operon encodes a putative sphingomyelin phosphodiesterase. This enzyme is a hydrolase involved in the metabolism of sphingolipids (*Goñi and Alonso, 2002*) and is a putative virulence factor (*Oda et al., 2012*). It is tempting to hypothesize that the homologues of VtrA/VtrC in these bacteria function as a sensor for sphingolipids and regulate the induction of the downstream gene that produces sphingomyelin phosphodiesterase.

In conclusion, using microbial genetics and biophysics, we have identified and characterized a bile salt sensor that is widely distributed and used to induce virulence by many *Vibrio*s. Additionally, we have found that a family of monomeric lipid binding calycin domain proteins has expanded to include an obligate heterodimer that binds bile salts and can be used to transmit a signal. To our knowledge, our study provides the first biochemical and structural analysis of a prokaryotic receptor involved in mediating a response to bile salts, a significant environmental cue during infection.

## Materials and methods

### Bacterial strains and cell culture

The *V. parahaemolyticus* clinical RimD2210633 derivative strain POR1 (Δ*tdhAS*) and its derivative strains were cultured in Marine LB (MLB) broth (LB broth containing 3% NaCl) or Marine minimal medium (MMM) at 30°C. *E.coli* DH5α, *E.coli* S17 (λ pir), B834(DE3), Rosetta 2(DE3), and BL21(DE3) were cultured in LB broth at 37°C. The medium was supplemented with kanamycin (30 µg/ml for *E. coli* and 240 µg/ml for *V. parahaemolyticus*) or chloramphenicol (25 µg/ml) when necessary.

### Plasmid construction

For vector induced expression of VtrC that is regulated by its native promoter, the 1kb upstream of *vtrA* followed by the *vtrC* coding sequence was PCR amplified and cloned into the pBAD/*Myc*-His vector (Invitrogen) in which ampicillin resistance was replaced with kanamycin resistance. To

generate pBAD-FLAG-*vtrC*ΔN30, *vtrC*ΔN30 (aa 31–161) was amplified by PCR (a FLAG epitope was added via the forward primer) and cloned into pBAD/Myc-His under the control of the arabinose inducible promoter. Amino acid residues 1 to 30 are predicted to encode the N-terminal transmembrane region. Since membrane localization is predicted to be important for the activity of VtrC, the VtrCΔN30 mutant protein should localize to the cytoplasm and, thus, be inactive. PhoA and LacZ fusions to VtrC in pBAD/Myc-His were generated by PCR amplifying phoA (amino acids 22–471, lacking the endogenous signal peptide) and the full length lacZ (amino acids 1–1024) and cloning directly upstream of VtrC aa 1 for N-terminal fusions or downstream of VtrC's last residue (aa 161) for C-terminal fusions.

## Deletion and knock-in strains

To generate the *vtrC* deletion, the 1kb upstream and downstream of the nucleotide sequence for *vtrC* gene (nucleotides 34–486) to be deleted were cloned into the pDM4, a Cm^ROriR6K suicide plasmid containing the *sacB* gene, which encodes an enzyme that metabolizes sucrose into a toxic product. C-terminal FLAG tag knock-in of *vopD2* and *vopS* were generated by cloning the 1kb upstream of the knock-in site, followed by the nucleotide sequence encoding the FLAG tag and the 1kb downstream of the knock-in site into pDM4. The resulting pDM4 plasmids were conjugated into the POR1 strain from *E. coli* S17 (λ pir) and the transconjugants were selected on medium containing 25 µg/ml chloramphenicol. Bacteria were then counterselected by growing on medium containing 15% sucrose to select for clones were the suicide plasmid had recombined out of the chromosome. PCR analysis was performed to confirm successful deletions and knock-ins and that the *vtrA* gene remained intact.

## Antibodies

FLAG antibodies were purchased from Sigma-Aldrich (F3165 and F7425). Polyclonal antibodies were produced in house with rabbits for the recombinant proteins for VopA and VopC and for the VtrA peptide aa 235–253.

## Bioinformatics

In order to identify VtrA and VtrC homologues, we performed PSI-BLAST (*Altschul et al., 1997*) searches against a non-redundant (NR) database (threshold E-value cutoff 0.02) with query sequences corresponding to the predicted periplasmic domains of VtrA (VPA1332: gi|28901187, aa 150–253) and VtrC (VPA1333: gi|28901188, aa 27–161). Because our VtrC PSI-BLAST hits were from different species than our VtrA hits, we initiated a transitive PSI-BLAST search against the NR database (threshold E-value cutoff 0.01) using the most distant identified VtrA sequence as a query (gi|494726765, aa 160–272). The transitive search found the VtrA sequences from the remaining species containing VtrC. All identified VtrA and VtrC hits were found in tandem in their respective genomes, and representative sequences from non-redundant species were collected. VtrB sequences were identified by searching VtrA/VtrC-containing genomes for the VtrB sequence (VPA1348: gi|28901203). Due to widespread nature and strong conservation of the VtrB helix-turn-helix sequence in many transcription factors, we required VtrB orthologs to: 1. be the top-scoring hit in the respective genomes, 2. retain the predicted C-terminal transmembrane helix and 3. be in close proximity to the VtrA/VtrC gene pair. Representatives from each sequence family were aligned using the MAFFT server (http://mafft.cbrc.jp/alignment/server/).

To calculate the structure-based distance tree, representative structures were chosen from the lipocalins/streptavidin group in the Evolutionary Classification Of protein Domain structures (ECOD) database. We chose structures with bound ligands, when available, from the main branches of the calycin superfamily (*Flower et al., 2000*): lipocalins (1kt7:1–175, 2aco:10–177), triabin (4n7c:3–176), fatty acid binding proteins (FABPs) (1o1v: 1–126, 3wvm:0–132), streptavidin (2f01:14–134), and metalloprotease inhibitor (MPI) (1jiw:1–105, chain I). We also included the HRI1 N-terminal domain structure (HRIN-like; 3rby:1–151) with a unique binding site that traverses the lipocalin-like barrel core, as well as the VtrC structure. The representative lipocalin-like domains were compared pairwise all-against-all using DaliLite (*Holm and Park, 2000*). Pairwise DaliLite Z-scores ($Z_{AB}$) were transformed to distances by comparing to self DaliLite Z-scores ($Z_{AA}$, $Z_{BB}$) using the following equation: $-\ln(Z_{AB}/$

$(0.5*Z_{AA}+0.5*Z_{BB}))$. The structure-based tree was produced using the FITCH program (with global optimization) of the Phylip package (*Felsenstein, 1997*).

## Quantitative RT-PCR

Total RNA was extracted from *V. parahaemolyticus* strains using RNeasy Plus Mini Kit (QIAGEN 74134). Extracted RNA was then reverse transcribed into cDNA using ProtoScript First Strand cDNA Synthesis Kit (NEB E6300S) utilizing Random Primer Mix. The resulting cDNA served as the template for quantitative RT-PCR analysis using iTaq Universal SYBR Green Supermix (Bio-Rad) and the ViiA7 Real-Time PCR System (Applied Biosystems). The $2^{-\Delta\Delta CT}$ method was used to determine the mRNA level of *vtrA* and *vtrB* in each sample relative to POR1 grown in LB without bile salts. The expression of *vtrA* and *vtrB* was normalized against the expression of *fliA*. Primers for specific genes were as follows: *vtrA*, 5'-TTGGAACCCACGAACATCTC-3' and 5'-CAGTCACAAATTTTCCTGGCC-3'; *vtrB*, 5'-ATTATCAGCTTAGGTGGGCG-3' and 5'-ACTTTACCCCACACTTTGTCG-3'; control gene *fliA*, 5'-AAGCGATAACCTATGACCAGC-3' and 5'-TCCTCTACCTGAACACTCGG-3'. For the PCR that tests if *vtrA* and *vtrC* are cotranscribed, the cDNA served as the template and the primers used were: 5'-AATTGTTCCAGAAAGGCTCTATGTCATGCTTAATG'-3 and 5'- GTTTCATAAAAATGAACTGGTTGAAAAAAATTG-3'.

## T3SS2 expression and secretion assay

V. *parahaemolyticus* strains were grown overnight in MLB at 30°C. For experiments that involved vector induced expression of VtrC under its endogenous promoter, overnight cultures were diluted to $OD_{600 \, nm}$ = 0.3 in LB supplemented with 0.05% bile salts or 0.5 mM individual bile acid taurodeoxycholate (TDC), glycodeoxycholate (GDC), chenodeoxycholate (CDC) or cholate (CA) and induced for 4 hr at 37°C. For experiments that involved vector induced expression of VtrC under the arabinose inducible promoter of pBAD, overnight cultures were diluted to $OD_{600 \, nm}$ = 0.6 in MLB supplemented with 0.1% arabinose and induced for 3 hr at 30°C. The cultures were then diluted to $OD_{600nm}$ = 0.3 in LB supplemented with 0.1% arabinose and 0.05% bile salts and induced for 4 hr at 37°C. For the expression fraction (cell), the same $OD_{600 \, nm}$ of bacterial cultures were collected and cell pellets were resuspended in 2x protein sample buffer (100 mM Tris·HCl pH 6.8, 20% glycerol, 2% sodium dodecyl sulfate (SDS), 2% β-mercaptoethanol, 150 mM sodium hydroxide, bromophenol blue). For the secretion fraction (sup), bacterial culture supernatants were filtered and precipitated with deoxycholate and trichloroacetic acid (*Kimata et al., 2004*). Precipitated proteins were pelleted and washed with acetone and then resuspended in 2x protein sample buffer. Protein expression and secretion were detected by western blot analysis.

## T3SS1 expression and secretion assay

V. *parahaemolyticus* strains were grown overnight in MLB at 30°C. Overnight cultures were diluted to $OD_{600 \, nm}$ = 0.3 in LB or Dulbecco's Modified Eagle Medium (DMEM) and induced for 4 hr at 37°C. For the expression fraction (cell), the same $OD_{600 \, nm}$ of bacterial cultures were collected and cell pellets were resuspended in 2x protein sample buffer. For the secretion fraction (sup), bacterial culture supernatants were filtered and precipitated with deoxycholate and trichloroacetic acid (*Kimata et al., 2004*). Precipitated proteins were pelleted and washed with acetone and then resuspended in 2x protein sample buffer. Protein expression and secretion were detected by western blot analysis.

## Alkaline phosphatase PhoA assay

V. *parahaemolyticus* strains containing the empty pBAD vector, pBAD-*phoA-vtrC,* or pBAD-*vtrC-phoA* were grown overnight in MLB at 30°C. Overnight cultures were diluted to $OD_{600 \, nm}$ = 0.6 in MLB supplemented with 0.1% arabinose and induced for 5 hr at 30°C. 1 ml of bacterial culture was collected, washed once, and resuspended with 1 ml Tris solution (1M Tris·HCl, pH 8.0). 50 µl of resuspended cells were transfered to 1 ml Tris solution and permeabilized by adding 30 µL 0.1% SDS and vortexing for 10 s. The mixture was then incubated at 37°C for 5 min. 100 µl of p--nitrophenyl phosphate (PNPP, Thermo Scientific Pierce 37621) was added to the samples to start the reaction and the time was recorded as T1 (min). Samples were incubated at 37°C until a pale yellow color develops. The reaction was stopped by adding 100 µl of cold 1 M $KH_2PO_4$ and the time

was recorded as T2 (min). The supernatant was collected after centrifugation at 20000 x g for 5 min. Measure the absorbance of the supernatant with a spectrophotometer at wavelength 420nm. Alkaline phosphatase activity was calculated as below:

Alkaline phosphatase activity (Miller unit) = 1000* $OD_{420\ nm}$/0.05* $OD_{600\ nm}$ *(T2-T1)

## β-galactosidase LacZ assay

V. *parahaemolyticus* strains containing the empty pBAD vector, pBAD-*lacZ-vtrC* or pBAD-*vtrC-lacZ* were grown overnight in MLB at 30°C. Overnight cultures were diluted to $OD_{600\ nm}$ = 0.6 in MLB supplemented with 0.1% arabinose and induced for 5 hr at 30°C. Bacterial cells were permeablized by mixing 20 µl of bacterial culture with 80 µl permeabilization solution (100 mM $Na_2HPO_4$, 20 mM KCl, 2 mM $MgSO_4$, 0.8 mg/ml hexadecyltrimethylammonium bromide, 0.4 mg/ml sodium deoxycholate, 5.4 µl/ml β-mercaptoethanol). 600 µl substrate solution (60 mM $Na_2HPO_4$, 40 mM NaH2PO, 1 mg/ml 2-Nitrophenyl β-D-galactopyranoside (ONPG, Sigma-Aldrich N1127), 2.7 µl/ml β-mercaptoethanol) was added to the samples to start the reaction and the time was recorded as T1 (min). Samples were incubated at 30°C until a pale yellow color develops. The reaction was stopped by adding 100 µl 1 M $Na_2CO_3$ and the time was recorded as T2 (min). The supernatant was collected after centrifugation at 20000 x g for 5 min. Measure the $OD_{420nm}$ of the supernatant. β-galactosidase activity was calculated as below:

β-galactosidase activity (Miller unit) = 1000* $OD_{420\ nm}$/0.02* $OD_{600\ nm}$ *(T2-T1)

## Immunoprecipitation

The following strains were used for the Co-IP experiments on *Figure 5C*: V. *parahaemolyticus* strains POR1 + pBAD, POR1Δ*vtrA* + pBAD-FLAG-*vtrC* and POR1 + pBAD-FLAG-*vtrC*. The following strains were used for the Co-IP experiments on *Figure 9C*: V. *parahaemolyticus* strains POR1 + pBAD, POR1Δ*vtrA* + pBAD-FLAG-*vtrC* and POR1 Δ*vtrC* + pBAD-FLAG-*vtrC* wild type and mutants H50R, Y81A and Q42A. Strains were grown overnight in MLB at 30°C. Overnight cultures were diluted to $OD_{600\ nm}$ = 0.6 in MLB supplemented with 0.1% arabinose and induced for 3 hr at 30°C. The cultures were then diluted to $OD_{600\ nm}$ = 0.3 in LB supplemented with 0.1% arabinose and 0.05% bile salts and induced for 4 hr at 37°C. 200 ml bacterial culture were resuspended with lysis buffer (50 mM Tris·HCl pH 7.4, 100 mM NaCl, 0.2% Triton X-100, 5 mM EDTA, 1 mg/ml lysozyme and 1 mM PMSF). Resuspended samples were shaken at 22°C for 30 min and then subjected to three freeze/thaw cycles. The supernatant was collected after centrifugation at 25,000 x g, 4°C for 30 min. Immunoprecipitation was performed by incubating the supernatant with Anti-FLAG M2 beads at 4°C with gentle shaking for 4 hr. Beads were collected and washed with wash buffer (50 mM Tris·HCl pH 7.4, 100 mM NaCl, 0.2% Triton X-100) 3 times. Proteins bound to beads were eluted with 2x protein sample buffer. VtrA and VtrC were detected by western blot analysis using Anti-VtrA and Anti-FLAG antibodies.

## Protein expression and purification

The periplasmic domain of VtrA (aa 161–253) was cloned as an N-terminally hexahistidine and maltose binding protein-tagged fusion protein into pET28b (Novagen) to produce pET28b-HisMBP-VtrA, which contains a Tobacco Etch Virus (TEV) protease cleavage site between the HisMBP-tag and VtrA. The pET28b-HisMBP-VtrA construct was expressed in *E. coli* Rosetta 2(DE3) cells (Novagen). For VtrC/VtrA coexpression, the periplasmic domains of VtrC (aa 31–161) and VtrA (aa 161–253) were cloned into the first and second multiple cloning site, respectively, of pACYCDuet-1 (Novagen) to produce pACYCDuet-VtrC/VtrA, where VtrC is N-terminally hexahistidine-tagged. The pACYCDuet-VtrC/VtrA construct was expressed in *E. coli* BL21(DE3) cells. All cultures were grown in LB at 37°C until $OD_{600nm}$ 0.5–0.6 and induced with 0.4 mM isopropyl β-D-thiogalactopyranoside (IPTG) overnight at 22°C for pET28b-HisMBP-VtrA, and 17°C for pACYCDuet-VtrC/VtrA. Selenomethionyl-derivatized VtrA/VtrC complex was expressed in *E. coli* B834(DE3) cells (Novagen) in SelenoMet medium (Molecular Dimensions). Cells were harvested by centrifugation, resuspended in buffer A (50 mM Tris pH 8.0 and 100 mM NaCl), and lysed by extrusion. Lysates were clarified by centrifugation and filtered (0.45-µm pore size). All proteins were purified by nickel-affinity purification using HisPur Ni-NTA resin (ThermoFisher, Rockford, Illinois) on a gravity flow column. Briefly, lysates were incubated with the resin for 30 min at 4°C with nutation. Lysate and beads were applied to the

column and washed with 20 column volumes of buffer A supplemented with 15 mM imidazole. Proteins were eluted with 5 column volumes of buffer B (50 mM Tris pH 8.0, 100 mM NaCl and 250 mM imidazole). For VtrA, the HisMBP fusion domain was removed by treating with TEV protease overnight at 4°C, followed by a second round of nickel-affinity chromatography. Proteins were further purified by size exclusion chromatography (SEC) on a Superdex 75 16/600 column (Pharmacia Biotech) with buffer A. All proteins were buffer exchanged to 10 mM Tris pH 8 and 10 mM NaCl for crystallographic studies. Selenomethionyl-derivatized VtrA/VtrC complex was purified in a similar manner with the addition of 1 mM DTT to all buffers.

## Gel filtration assay

Gel filtration was performed on a Superdex S75 16/600 column (Pharmacia Biotech) with buffer containing 50 mM Tris pH 8.0 and 100 mM NaCl at a flow rate of 1 ml/min. The protein complex solution was injected into the column at a final concentration of 1 mg/ml in a total volume of 1 ml. Fractions were analyzed by SDS-PAGE and Coomassie blue staining. For protein molecular weight determination, the column was calibrated with a gel filtration LMW calibration kit (GE Healthcare) using the running buffer described above.

## Crystallization and X-ray data collection

Crystals of native VtrA/VtrC periplasmic domain heterodimer were grown using the sitting-drop vapor diffusion method from drops containing 0.2 μl protein (13 mg/ml) and 0.2 μl reservoir solution (1.0 M lithium sulfate, 0.5 M ammonium sulfate, 0.1 M sodium citrate pH 5.6) and equilibrated over 50 μl reservoir solution at 20°C. Crystals of selenomethionyl-derivatized heterodimer were grown using the hanging-drop vapor diffusion method from drops containing 1 μl protein (10 mg/ml) and 1 μl of reservoir solution (1.0 M lithium sulfate, 0.5 M ammonium sulfate, 0.1 M sodium citrate pH 5.9) and equilibrated over 500 μl of reservoir solution. Crystals appeared after 2 days at 20°C and grew to their maximal extent by 1 week. Native and selenomethionyl-derivatized crystals were cryoprotected by transferring to a final solution of 28.7% ethylene glycol, 1.1 M lithium sulfate, 0.6 M ammonium sulfate, and 0.1 M sodium citrate pH 5.6–5.8, then the crystals were flash-cooled in liquid nitrogen. Crystals of the VtrA/VtrC heterodimer bound to the bile salt TDC were grown with protein that had been loaded with TDC by performing purification in the presence of 0.5 mM TDC in all buffers and buffer exchanging into 10 mM Tris pH 8, 10 mM NaCl. Drops containing 1 μl protein (5 mg/ml) and 1 μl of reservoir solution (2.0 M ammonium sulfate, 0.1 M sodium acetate pH 4.6) were set up by the hanging-drop vapor diffusion method, and equilibrated over 500 μl of reservoir solution. Crystals appeared after 6 days at 20°C and grew to their maximal extent by 2 weeks. Crystals were cryoprotected by transferring to a final solution of 22.5% ethylene glycol, 2.1 M ammonium sulfate, 0.1 M sodium acetate pH 4.6, and 0.5 mM TDC, and flash-cooled in liquid nitrogen.

Data were collected at APS beamline 19-ID at 100 K, and were indexed, integrated and scaled using the HKL-3000 program package (*Minor et al., 2006*). Native and selenomethionyl-derivatized VtrA/VtrC heterodimer crystals exhibited the symmetry of space group F432 with unit cell parameters of a = 211.01 Å and contained one molecule each of VtrA and VtrC per asymmetric unit, with a solvent content of 65%. TDC-containing crystals belonged to space group $P2_12_12_1$ with unit cell parameters of a = 55.39 Å, b = 71.28 Å and c = 203.73 Å and contained three molecules each of VtrA/VtrC heterodimer per asymmetric unit, with a solvent content of 50%. Native, selenomethionyl-derivatized and TDC-containing crystals diffracted isotropically to a $d_{min}$ of 2.70 Å, 2.60 Å and 2.10 Å, respectively, when exposed to synchrotron radiation. Data collection statistics are provided in *Table 1*.

## Phase determination and structure refinement

Phases for the native VtrA/VtrC heterodimer were obtained from a two-wavelength anomalous dispersion experiment using selenomethionyl-derivatized heterodimer protein with data to a $d_{min}$ of 2.60 Å. Two selenium sites were located using the program *SHELXD* (*Schneider and Sheldrick, 2002*), and phases were refined with the program *Mlphare* (*Otwinowski, 1991*), resulting in an overall figure-of-merit of 0.25 for data between 121.8 and 2.60 Å. Phases were further improved by density modification in the program *Parrot* (*Cowtan, 2010*) resulting in a figure-of-merit of 0.88. An

**Table 1.** Data collection and refinement statistics, VtrC/VtrA complex.

**Data collection**

| Crystal | SeMetpeak[*] | SeMet inflection point[*] | Native | Native + bile salt |
|---|---|---|---|---|
| Space group | F432 | F432 | F432 | P2₁2₁2₁ |
| Cell constants (Å) | a = 211.01 | a = 211.46 | a = 211.39 | a = 55.39, b = 71.28, c = 203.73 |
| Wavelength (Å) | 0.97927 | 0.97943 | 0.97935 | 0.97926 |
| Resolution range (Å) | 35.67–2.60 (2.64–2.60) | 35.74–2.65 (2.70–2.65) | 40.68–2.65 (2.70–2.65) | 35.64–2.10 (2.14–2.10) |
| Unique reflections | 12,839 (620) | 12,267 (593) | 12,322 (594) | 45,762 (1868) |
| Multiplicity | 22.6 (20.2) | 22.7 (21.8) | 36.1 (36.5) | 3.9 (2.8) |
| Data completeness (%) | 99.9 (100.0) | 99.9 (100.0) | 99.9 (100.0) | 95.6 (79.4) |
| $R_{merge}$ (%)[†] | 7.2 (369.2) | 7.7 (292.5) | 7.6 (172.4) | 7.9 (36.2) |
| $R_{pim}$ (%)[‡] | 1.5 (83.7) | 1.7 (63.5) | 1.3 (28.8) | 4.3 (24.9) |
| $CC_{1/2}$ (last resolution shell) | 0.448 | 0.520 | 0.934 | 0.754 |
| I/σ(I) | 49.9 (0.8) | 46.8 (1.1) | 63.6 (4.2) | 15.9 (2.5) |
| Wilson $B$-value (Å²) | 79.0 | 78.6 | 77.6 | 31.1 |
| Wilson $B$-value, sharpened (Å²)[§] | 35.8 | 35.4 | 37.5 | 24.2 |

**Phase determination**

| | |
|---|---|
| Anomalous scatterers | selenium, 2 out of 5 possible sites |
| Figure of merit (121.8–2.60 Å) | 0.25 after Selenium MAD phasing; 0.88 after density modification |

**Refinement statistics**

| | | | | |
|---|---|---|---|---|
| Resolution range (Å) | | | 40.68–2.70 (2.79–2.70) | 35.64–2.10 (2.16–2.10) |
| No. of reflections $R_{work}$/$R_{free}$ | | | 11,593/1,470 (899/129) | 41,304/2,000 (1,807/91) |
| Data completeness (%) | | | 99.9 (100.0) | 86.3 (57.0) |
| Atoms (non-H protein/ions/ligands/solvent) | | | 1,797/5/NA/NA | 5,411/NA/375/204 |
| $R_{work}$ (%) | | | 26.0 (34.0) | 20.1 (22.3) |
| $R_{free}$ (%) | | | 29.8 (39.2) | 23.3 (28.3) |
| R.m.s.d. bond length (Å) | | | 0.002 | 0.002 |
| R.m.s.d. bond angle (°) | | | 0.42 | 0.51 |
| Mean $B$-value (Å²) (protein chain ID) (ligands/ions/solvent) | | | VtrA(A): 77.2 VtrC(B): 62.2. ions: 70.2 | VtrA(A): 25.7 VtrC(B): 29.8 VtrA(C): 34.9 VtrC(D): 37.8 VtrA(E): 39.1 VtrC(F): 36.2 ligands: 43.3 ions: 86.2 solvent: 37.0 |
| Ramachandran plot (%) (favored/additional/disallowed)[#] | | | 94.3/5.2/0.5 | 96.5/3.2/0.3 |
| Maximum likelihood coordinate error | | | 0.36 | 0.23 |
| Missing residues, protein (chain ID) | | | VtrA(A): 161– 163. VtrC(B): −13 –0, 113–117. | VtrA(A): 161–164. VtrC(B): −13–0. VtrA(C): 161–163. VtrC(D): 13–0, 119–124. VtrA(E): 161–165. VtrC(F): 13–0. |

Data for the outermost shell are given in parentheses.

*Bijvoet-pairs were kept separate for data processing.

[†]$R_{merge} = 100 \sum_h \sum_i | I_{h,i} - \langle I_h \rangle | / \sum_h \sum_i \langle I_{h,i} \rangle$, where the outer sum ($h$) is over the unique reflections and the inner sum ($ii$) is over the set of independent observations of each unique reflection.

[‡]$R_{pim}$ is the precision indicating $R$-factor, i.e $R_{pim} = 100 \sum_h \sum_i [1/(n_h - 1)]^{1/2} \mid I_{h,i} - \langle I_h \rangle \mid / \sum_h \sum_i \langle I_{h,i} \rangle$, where $n_h$ is the number of observations of reflections $h$ (**Evans, 2011**).
[§]$B$-factor sharpening was performed in the autocorrection mode of *HKL3000* (**Borek et al., 2013**).
[#]Asdefined by the validation suite MolProbity (**Chen et al., 2010**).

initial model containing 84% of all VtrA/VtrC heterodimer residues was automatically generated in the program *Buccaneer* (**Cowtan, 2006**).

As the selenomethionyl-derivatized and native crystals were isomorphous, all further calculations for the native structure were performed versus the native data. Additional residues for the VtrA/VtrC heterodimer were manually modeled in the program *Coot* (**Emsley et al., 2010**). Positional and isotropic atomic displacement parameter (ADP) as well as TLS ADP refinement was performed to a resolution of 2.70 Å using the program *Phenix* (**Afonine et al., 2010**) with a random 10% of all data set aside for an $R_{free}$ calculation. The current model contains one VtrA/VtrC heterodimer; included are residues residues 164–253 of VtrA 31–112 and 118–161 of VtrC, and one sulfate ion. The $R_{work}$ is 0.260, and the $R_{free}$ is 0.298. A Ramachandran plot generated with *Molprobity* (**Chen et al., 2010**) indicated that 94.3% of all protein residues are in the most favored regions and 0.5% (one residue) in disallowed regions.

Phases for the TDC-containing heterodimer were obtained by the molecular replacement method in the program *Phaser* (**McCoy et al., 2007**) using the coordinates for the native VtrA/VtrC heterodimer. Model building and refinement were performed to a resolution of 2.1 Å using a similar protocol to the native structure. Four TDC molecules were located in the asymmetric unit, one each bound to the VtrC monomer of a VtrA/VtrC heterodimer and one at a lattice contact between VtrA molecules. The current model contains three VtrA/VtrC heterodimers in the asymmetric unit, four TDC molecules, seven sulfate ions and 204 water molecules. A *Molprobity* (**Chen et al., 2010**) generated Ramachandran plot indicates that 96.5% of all protein residues are in the most favored regions and 0.3% (two residues) in disallowed regions. Phasing and model refinement statistics for all structures are provided in *Table 1*.

## Isothermal titration calorimetry (ITC)

The VtrA/VtrC periplasmic domain complex was dialyzed at 4°C overnight against the assay buffer (50 mM Tris pH8, 100 mM NaCl). Taurodeoxycholic acid (400 µM) was prepared by dissolving the dry powder (Sigma) with the same dialysis buffer. ITC experiments were performed at 25°C on a MicroCal iTC200 system (Malvern), with reference power at 5 µcal/s and stirring rate at 750 rpm. Measurements were performed as 19 injections of 400 µM taurodeoxycholic acid (1 µl for the first injection and 2 µl for injections 2–19) into approximately 200 µl of 36 µM VtrA/VtrC. ITC data were integrated and analyzed using NITPIC 1.1.5 (**Keller et al., 2012**; *Scheuermann and Brautigam, 2015*) and ITCsy version 1a (*Brautigam et al., 2016*). ITC data plots were prepared with GUSSI 1.1.0 (*Brautigam, 2015*).

## Acknowledgements

We thank the Orth Lab for valuable advice and helpful discussions. We gratefully acknowledge Dr. Chad Brautigam and Dr. Thomas Scheuermann for assistance with ITC data collection and Dr. Zhe Chen for assistance with X-ray data collection. The structure in this report is derived from work performed on beamline 19-ID at the Argonne National Laboratory, Structural Biology Center at the Advanced Photon Source, operated by UChicago Argonne, LLC, for the US Department of Energy, Office of Biological and Environmental Research under contract DE-AC02-06CH11357. This work was supported by NIH grant R01-AI087808, (KO) and R01-GM094575 (NVG), Welch Research Foundation Grants I-1561 (KO) and I-1505 (NVG).

## Additional information

### Funding

| Funder | Grant reference number | Author |
|---|---|---|
| National Institutes of Health | K99AI116948 | Dor Salomon |
| U.S. Department of Energy | AC02-06CH11357 | Diana R Tomchick |
| National Institutes of Health | R01-GM094575 | Nick V Grishin |
| Welch Foundation | I-1505 | Nick V Grishin |
| National Institutes of Health | R01-AI087808 | Kim Orth |
| Welch Foundation | I-1561 | Kim Orth |

The funders had no role in study design, data collection and interpretation, or the decision to submit the work for publication.

### Author contributions

PL, GR-C, LNK, Conception and design, Acquisition of data, Analysis and interpretation of data, Drafting or revising the article; DS, NVG, KO, Conception and design, Analysis and interpretation of data, Drafting or revising the article; DRT, Conception and design, Acquisition of data, Analysis and interpretation of data, Drafting or revising the article, Contributed unpublished essential data or reagents

### Author ORCIDs

Diana R Tomchick, http://orcid.org/0000-0002-7529-4643
Kim Orth, http://orcid.org/0000-0002-0678-7620

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
