## [Decision Letter]

[Editors’ note: this article was originally rejected after discussions between the reviewers, but the authors were invited to resubmit after an appeal against the decision.]

Thank you for submitting your work entitled "Bile Salt Receptor Complex Activates a Pathogenic Type III Secretion System" for consideration by *eLife*. Your article has been reviewed by two peer reviewers, and the evaluation has been overseen by a Reviewing Editor and Richard Losick as the Senior Editor. The following individuals involved in review of your submission have agreed to reveal their identity: Alejandro Buschiazzo (Reviewer #2).

Our decision has been reached after consultation between the reviewers. Based on these discussions and the individual reviews below, we regret to inform you that your work will not be considered further for publication in *eLife*.

The reviewers and editor found the work intriguing. However, there was unanimity in the feeling that the manuscript did not provide an understanding of bile recognition in the investigated signal transduction pathway. Moreover, there were several technical and conceptual issues that are detailed in the comments presented below.

Reviewer #1:

Li P et al. report the discovery of a new protein (VtrC) as an essential element to regulate type three secretion system 2 in Vibrio. VtrC is shown to form a complex with VtrA, which is responsive to bile (a known inductor of T3SS2, ultimately modulating virulence). VtrA and VtrB had already been identified as essential T3SS2 regulators, but the mechanism of bile sensing and signal transduction remains unknown, and this is the main question the authors want to address.

After identifying VtrC, the authors demonstrate that vtrA and vtrC belong to a single operon and, elegantly, that deletion of vtrC specifically affects T3SS2 and bile sensitivity.

VtrC is further characterized as a bitopic membrane protein, which led to uncovering periplasmic interaction with VtrA in vivo (co-IP). Not affecting vtrA transcription, VtrC appears to be critical for VtrA protein stability.

1) Why didn't the authors raise an antibody against the whole periplasmic (or intracellular) domain of VtrA? (VtrA might be expressed in the absence of VtrC, but that last short segment is being degraded, or that epitope is conformational and sensitive to VtrC binding?)

2) The crystal structure of the VtrA/VtrC complex represents an important contribution, confirming the association between both proteins.

Some concerns about the crystallographic data (the authors' feedback is important):

• for a 2-wavelength MAD phasing, 0.25 seems a suspiciously low FOM figure (what can you say about the three Se sites that were not found with direct methods, on the final refined model?)

• Rfree in principle seems high (almost 30%) for a 2.7Å resolution structure. No electron density is shown. Were there any difficulties throughout refinement? Is electron density good throughout the asymmetric unit? (please include a supplementary figure with best and worst ED map regions)

• Stereochemical rmsd figures for bond lengths and angles are extremely low. Why? Were extra-tight restraints used during refinement? if yes, why?

Please briefly address these points in the Materials and methods.

3) Are there other, alternative, heterodimer architectures (or yet, tetrameric organizations etc) in the cubic packing? Please cite the next best inter-protomer buried interface surface, ruling out otherwise ambiguous oligomeric choices.

4) The structural data don't seem to contribute all that much into a mechanistic understanding. Particularly, the ligand-binding site cannot be pinpointed as the authors admit. Did the authors attempt crystallizing the complex in the presence of TDC or other potential ligands? Make this explicit. Structurally assessing how VtrA-VtrC recognizes bile would be a major mechanistic insight (see next).

5) TDC is used as the sole ligand assayed by ITC, based on a previous report, but bile is really heterogeneous. The authors would benefit of incubating the complex with bile, then purifying and crystallizing (also MS can be helpful), attempting to see and/or identify ligands, which may not be TDC. Actually, the authors claim that TDC binds with nM affinity, which is somewhat misleading. KD is actually on the low microM range (i.e. nearer microM than nM, in contrast to most FABPs used as reference in Richieri et al. 2000 Biochemistry). "Nanomolar" should thus be changed to "low μM" in the Abstract and Discussion. Maybe other bile components bind tighter and are relevant physiologically?

6) Structural data would also be valuable to understand the critical role of interacting residues in the VtrA/VtrC interface. Structure-based mutants of vtrC can be analyzed on the vtrC-knockout background by the kind of functional assays that the authors have performed.

7) The major concern overall for the paper is indeed the mechanistic insight, which was the main question the authors sought to answer in the first place. Figure 9 is too vague: which bile component(s) binds physiologically? If it were TDC, where does it bind on the VtrA/VtrC complex? What are the characteristics of the signal-triggered rearrangement of VtrA (and/or VtrC)?

Alternative hypothesis: VtrC stabilizes (somehow) VtrA, but it is VtrA on its own that binds bile components and transduces the signal. Given that recombinant VtrA's periplasmic domain is available, it would be important to assess its own capacity to bind TDC (using ITC). And, if really no binding is detectable, did the authors attempt crystallizing the VtrA periplasmic domain on its own? This structure could well inform about VtrC-triggered rearrangements.

Reviewer #2:

This manuscript by Li et al. describes research that increases understanding of how bile induces Type Three Secretion in Vibrio parahaemolyticus. It had previously been shown that the proteins VtrA and VtrB were necessary for bile-mediated induction of the T3SS2 system, and VtrA activated VtrB in response to bile. This paper identifies a new component of this system, VtrC, that is in the membrane associated with VtrA. The authors crystalize the periplasmic interacting domains of VtrA and VtrC, in the absence of bile, and show the complex binds bile. This is a nice manuscript that definitely increases our understanding of this process, but I feel the authors overstep a bit in terms of the novelty of this work and their conclusions that they provide mechanistic insight into this process. Therefore, my enthusiasm is somewhat tempered.

1) The authors state that "the mechanism of bile sensing by these bacteria remains elusive". From the introduction, it would seem that not bile responsive proteins have been identified, or the impact of bile on these proteins elucidated. However, in Vibrio cholerae ToxT binding to bile inhibits its ability to bind target promoters, and as the authors state in the discussion bile has been shown to induce disulfide bonds in ToxR.

2) The authors don't really uncover the mechanism of VtrAC bile sensing, but just another important component of the system. For instance, I was surprised that the authors did not use the crystal structure to engineer mutations that would be predicted to disrupt the interaction or disrupt the binding to bile. Moreover, no structures are determined with bile, and they only identify the structure of the periplasmic domains, so how bile binding is transduced to changes in DNA binding affinity is not understood. They clearly make important advances, but this manuscript does not provide the mechanism of bile sensing.

[Editors’ note: what now follows is the decision letter after the authors submitted for further consideration.]

Thank you for resubmitting your work entitled "Bile Salt Receptor Complex Activates a Pathogenic Type III Secretion System" for further consideration at *eLife*. Your revised article has been favorably evaluated by Richard Losick (Senior editor), a Reviewing editor, and two reviewers.

The manuscript has been improved but there are some remaining issues that need to be addressed before acceptance, as outlined below.

There some questions about the processing of the diffraction data, which should be addressed before the manuscript can be accepted for publication. These questions are as follows:

The authors should be aware that the F432 cubic space group is extremely rare in protein crystal structures (~200 entries belonging to it, among the >10^5^ structures in the PDB). So, although possible to encounter, as with any unusual discovery, one should be extra cautious to avoid mistakes.

For the next few comments, you might want to check whether your space group assignment was correct (or could there be a form of pseudo-symmetry?). Pseudo-symmetry often results in regions of the alleged ASU with weak electron density and greater difficulty in model building. This is why I had asked previously about how good the electron density is, not only in the good parts, but also in the bad ones (the authors misunderstood replying only with the obvious: that density is better in the core and worse in the flexible solvent-exposed regions). A mistake in the space group could end up being worrisome in giving parts of the model wrong (even on top of parts that are right). The new structure in complex with TDC, is in part reassuring (a confirmation with different packing), but some figures of its own, also raise questions.

Issues that have remained unanswered:

1) Phasing FOM too low: 0.25 is suspicious for a MAD experiment. Could be explained because of too little anomalous signal (e.g. only two methionines for approx. 200 residues – probably a quite tough phasing scenario). Again, my point was to think (and test!) whether there is a lower symmetry possibility to integrate the data in, and then see if shelxd works better. Straightforward tests that you can also perform to scrutinize this SG assignment: a) use ccp4 program zanuda to analyze your cubic data, it will systematically test for lower compatible space group choices (and also perform some initial refinement for each, to have comparative figures); b) reintegrate the cubic data in P1, and perform mol replacement now that you have a solution (which is maybe completely right, or at least partially right for sure), searching for as many hetero-dimers as you calculate according to the triclinic unit cell volume: if a solution is found, than you can analyze if the packing indeed conforms to a such highly symmetric space group as F432.

2) Maybe it is cubic, end of story.

3) But, if alternative choices do appear, not only shelxd might give better FOM, but actually maps may turn out better (in the bad parts, if there were); final refined average B factors closer to Wilson B (this is a definitely strange behavior reported in your table); better *Rmerge* statistics (also high, not only in the high resolution shell, but also in the low res – I know high multiplicity is here at play, so I do not want to say this is necessarily wrong: just look closely and critically at it, to avoid mistakes).

4) I would strongly advise against including hydrogen atoms in the refinements. Else, explain really well why you think it's OK to include them. This goes also for the orthorhombic form with TDC. The validation reports seem clear in that you have included them: to support a refinement strategy with independent H atoms you would need ultra-high resolution (i.e. better than 1.2Å) or a neutron light source for the diffraction experiment.

5) The nominal resolution for the new orthorhombic structure is probably not 2.1Å: very low completeness of the data (57% high res shell). Please reconsider this. Actually, please elaborate as to why completeness drops significantly comparing the processed data vs those used for refinement.

It is suggested to reduce/simplify subsection “The VtrA/VtrC complex is an obligate heterodimer”; it's enough there to indicate that only 2 Se-Met out of the 5 present in the protein complex, were ultimately seen to be ordered in the refined model, hence contributing proportionally little to the measured anomalous signal in the first place (likely explaining the low FOM figure at the phasing step, previous to density modification).

In this context it is rather misleading to state that the 2 Se-Met "were used for refinement", actually you are probably talking here about their use to solve the substructure of anomalous scatterers for subsequent protein phasing purposes.

---

## [Author Response]

*Reviewer #1:*

*Li P et al. report the discovery of a new protein (VtrC) as an essential element to regulate type three secretion system 2 in Vibrio. VtrC is shown to form a complex with VtrA, which is responsive to bile (a known inductor of T3SS2, ultimately modulating virulence). VtrA and VtrB had already been identified as essential T3SS2 regulators, but the mechanism of bile sensing and signal transduction remains unknown, and this is the main question the authors want to address.*

*After identifying VtrC, the authors demonstrate that vtrA and vtrC belong to a single operon and, elegantly, that deletion of vtrC specifically affects T3SS2 and bile sensitivity.*

VtrC is further characterized as a bitopic membrane protein, which led to uncovering periplasmic interaction with VtrAin vivo (co-IP). Not affecting vtrA transcription, VtrC appears to be critical for VtrA protein stability.

1) Why didn't the authors raise an antibody against the whole periplasmic (or intracellular) domain of VtrA? (VtrA might be expressed in the absence of VtrC, but that last short segment is being degraded, or that epitope is conformational and sensitive to VtrC binding?)

We made a peptide antibody to the most antigenic region of VtrA found on its C-terminus and used this reagent in many of our studies. We agree with the reviewer that the possibility exists that the C-terminal fragment of VtrA could be degraded and we would miss this with our antibody. To avoid confusion, we have rephrased the following sentence "Taken together, VtrC appears to function upstream of VtrB through a mechanism that may involve stabilizing VtrA." to "Thus, VtrC appears to function upstream of VtrB via a signaling cascade including VtrA. The observed disappearance of VtrA in the absence of VtrC could support a direct physical interaction between VtrA and VtrC." This hypothesis is then validated by our structural studies with the VtrA/VtrC complex.

2) The crystal structure of the VtrA/VtrC complex represents an important contribution, confirming the association between both proteins.

Some concerns about the crystallographic data (the authors' feedback is important):

*• for a 2-wavelength MAD phasing, 0.25 seems a suspiciously low FOM figure (what can you say about the three Se sites that were not found with direct methods, on the final refined model?)*

We have included the following explanation in the Results:

“The structure was solved by multiple-wavelength anomalous dispersion phasing using anomalous signals from selenomethionine and refined to a resolution of 2.70 Å using native data from an isomorphous crystal. Two of the five methionines, VtrA Se-Met 210 at the interface with VtrC and VtrC Se-Met 49, were used for refinement. Another two methionines showed no electron density and we presume they are highly disordered because they are N-terminal methionines. The final missing Se-Met is Se-Met 115 from VtrC, which is in the disordered loop that covers the β-barrel.”

The quoted Figure of Merit (FOM) of 0.25 was calculated at the end of phasing and thus is not inflated due to assumptions made during density modification. The FOM after density modification was 0.88; this is clearly stated in the Materials and methods section, “Phase determination and structure refinement.

*• Rfree in principle seems high (almost 30%) for a 2.7Å resolution structure. No electron density is shown. Were there any difficulties throughout refinement? Is electron density good throughout the asymmetric unit? (please include a supplementary figure with best and worst ED map regions)*

Extensive efforts were made to appropriately model the native apo structure in the F432 space group. Several factors contribute to the R_free_ value observed in this structure, including the high solvent content (66%), the presence of a significant percentage of disordered and therefore unmodeled residues at the N-termini of both VtrA and VtrC as well as the disordered loop of VtrC (9% of all residues in the asymmetric unit), a lack of sidechain density for a significant number of residues on the solvent-exposed surface of the heterodimer, as well as significant differences in the mean B-values between VtrA and VtrC (62 Å2 vs. 77 Å2, respectively). Despite these issues, the observed R_free_ is within acceptable limits for the observed high resolution limit of the data (Kleywegt G.J. and Brünger A.T. (1996) Checking your imagination: applications of the free R value. *Structure***4**:897-904). The “best” electron density map regions correspond to the hydrophobic core of each monomer, while the “worst” regions correspond to the surface of the heterodimer that is exposed to the bulk solvent.

*• Stereochemical rmsd figures for bond lengths and angles are extremely low. Why? Were extra-tight restraints used during refinement? if yes, why?*

Please briefly address these points in the Materials and methods.

Standard methods were used to refine the model in PHENIX, including the automated optimization of the weights for the X-ray observations/stereochemistry and X-ray observations/ADP (atomic displacement parameters). This optimization balances the results of minimization of R_work_ and R_free_ with the stereochemical and ADP restraints, and prevents chemically unreasonable geometry distortions. Given the medium-low resolution of the dataset plus the high solvent content (as discussed in the previous point) as well as the excellent geometry of the final model, it appears that PHENIX has done an appropriate job in this case.

3) Are there other, alternative, heterodimer architectures (or yet, tetrameric organizations etc) in the cubic packing? Please cite the next best inter-protomer buried interface surface, ruling out otherwise ambiguous oligomeric choices.

An analysis of the macromolecular interfaces found in the native apo structure in the F432 space group by the web server PDBePISA (http://www.ebi.ac.uk/pdbe/pisa/pistart.html) (E. Krissinel and K. Henrick (2007). 'Inference of macromolecular assemblies from crystalline state.'. J. Mol. Biol. **372**, 774--797) was performed. Results indicate that the VtrA/VtrC heterodimer is the most stable quaternary structure in solution. The analysis yields a buried surface area of 2,300 Å2 between VtrA and VtrC, with ∆G^int^ = -9.6 kcal/mol (∆G^int^ is the free energy gain due to bringing the subunits together into a complex), ∆G^diss^ = 3.1 kcal/mol (∆G^diss^ is the free energy of dissociation), plus two salt bridges and ten hydrogen bonds at the interface. All other potential oligomeric assemblies, despite having similar or larger buried surface areas, had unfavorable (negative) values of ∆G^diss^ and were biologically incompatible with the requirement that all VtrA and VtrC proteins must be anchored into the same membrane by N-terminal helical segments on each protein. In addition, these alternative assemblies are not observed in the native bile-salt bound crystalline lattice, space group P2_1_2_1_2_1_.

*4) The structural data don't seem to contribute all that much into a mechanistic understanding. Particularly, the ligand-binding site cannot be pinpointed as the authors admit. Did the authors attempt crystallizing the complex in the presence of TDC or other potential ligands? Make this explicit. Structurally assessing how VtrA-VtrC recognizes bile would be a major mechanistic insight (see next).*

We have solved the structure of the VtrA.VtrC complex bound to TDC and this has indeed provided “major mechanistic insight”.

*5) TDC is used as the sole ligand assayed by ITC, based on a previous report, but bile is really heterogeneous. The authors would benefit of incubating the complex with bile, then purifying and crystallizing (also MS can be helpful), attempting to see and/or identify ligands, which may not be TDC.*

We have included a new supplemental figure, Figure 5—figure supplement 1, showing the activation of T3SS2 by specific bile salts. Previously, Gotoh et al. (2010) established that bile salts are the component of bile that activates the T3SS2. We have recapitulated this data, and shown that the bile salt TDC is an activator of this system. Therefore, we used TDC for our biophysical analysis of the VtrA/VtrC complex.

Actually, the authors claim that TDC binds with nM affinity, which is somewhat misleading. KD is actually on the low microM range (i.e. nearer microM than nM, in contrast to most FABPs used as reference in Richieri et al. 2000 Biochemistry). "Nanomolar" should thus be changed to "low μM" in the Abstract and Discussion. Maybe other bile components bind tighter and are relevant physiologically?

In Richieri's paper, they tested FABPS from different sources against a variety of fatty acids by the ADIFAB fluorescence method. Depending on the specific FABP and fatty acid combination, Kd values varied from 2-367 nM. The affinity of Vtra/VtrC for TDC is comparable with the higher end of this range. Previous work by Gotoh at al. and new supplemental Figure 5—figure supplement 1 demonstrate that TDC is a physiologically relevant molecule for this system.

6) Structural data would also be valuable to understand the critical role of interacting residues in the VtrA/VtrC interface. Structure-based mutants of vtrC can be analyzed on the vtrC-knockout background by the kind of functional assays that the authors have performed.

We have addressed this comment with experiments in Figure 9 that show changing amino acids within the ligand-binding pocket disrupts bile salt mediated signaling, as this is the focus of this paper. Future studies will include mutations that address signal transduction mediated by bile salt binding; however, these studies are clearly beyond the scope of this paper.

7) The major concern overall for the paper is indeed the mechanistic insight, which was the main question the authors sought to answer in the first place. Figure 9 is too vague: which bile component(s) binds physiologically? If it were TDC, where does it bind on the VtrA/VtrC complex? What are the characteristics of the signal-triggered rearrangement of VtrA (and/or VtrC)?

*Alternative hypothesis: VtrC stabilizes (somehow) VtrA, but it is VtrA on its own that binds bile components and transduces the signal. Given that recombinant VtrA's periplasmic domain is available, it would be important to assess its own capacity to bind TDC (using ITC). And, if really no binding is detectable, did the authors attempt crystallizing the VtrA periplasmic domain on its own? This structure could well inform about VtrC-triggered rearrangements.*

We have solved the co-crystal structure of VtrA/VtrC with bile salts. As predicted, based on calycin domains, the bile salt binds in the hydrophobic chamber of the VtrA/VtrC complex. Rearrangements in the structure when binding TDC are observed and discussed. Mutant experiments with residues in the hydrophobic chamber corroborate the finding that bile salts bind in the hydrophobic pocket

Parenthetically, early in our studies we tried to see if recombinantly purified VtrA could bind bile salts using ITC. We saw no binding (Figure 1). After solving the structures of the VtrA/VtrC complex +/- bile salts and realizing it exists as an obligate heterodimer, we saw no reason to include this negative data.

*Reviewer #3:*

*1) The authors state that "the mechanism of bile sensing by these bacteria remains elusive". From the introduction, it would seem that not bile responsive proteins have been identified, or the impact of bile on these proteins elucidated. However, in Vibrio cholerae ToxT binding to bile inhibits its ability to bind target promoters, and as the authors state in the discussion bile has been shown to induce disulfide bonds in ToxR.*

We have excluded the general statement “bile sensing by these bacteria remains elusive”. We have included studies that have implicated bile salts in ToxT signaling. However, receptors involved in sensing bile salts and activating T3SS2 were unknown before this study.

*2) The authors don't really uncover the mechanism of VtrAC bile sensing, but just another important component of the system. For instance, I was surprised that the authors did not use the crystal structure to engineer mutations that would be predicted to disrupt the interaction or disrupt the binding to bile. Moreover, no structures are determined with bile, and they only identify the structure of the periplasmic domains, so how bile binding is transduced to changes in DNA binding affinity is not understood. They clearly make important advances, but this manuscript does not provide the mechanism of bile sensing.*

We have solved the co-crystal structure of VtrA/VtrC with bile salts. As predicted, it binds in the hydrophobic chamber of the VtrA/VtrC complex. Mutant experiments that corroborate this observation are also included in Figure 9.

Showing how bile binding is transduced to changes in DNA binding affinity of a transcription factor is work that we will address in the future and is out of the scope of this manuscript.

[Editors’ note: what now follows is the decision letter after the authors submitted for further consideration.]

*There some questions about the processing of the diffraction data, which should be addressed before the manuscript can be accepted for publication. These questions are as follows:*

*The authors should be aware that the F432 cubic space group is extremely rare in protein crystal structures (~200 entries belonging to it, among the >10^5^ structures in the PDB). So, although possible to encounter, as with any unusual discovery, one should be extra cautious to avoid mistakes.*

*For the next few comments, you might want to check whether your space group assignment was correct (or could there be a form of pseudo-symmetry?). Pseudo-symmetry often results in regions of the alleged ASU with weak electron density and greater difficulty in model building. This is why I had asked previously about how good the electron density is, not only in the good parts, but also in the bad ones (the authors misunderstood replying only with the obvious: that density is better in the core and worse in the flexible solvent-exposed regions).*

The authors have now provided two figures, Figure 11 and Figure 12 illustrating good versus poor electron density in the structure, as exemplified by the lack of density for the high percentage of solvent-exposed side chains (solvent content of the lattice is 68%).

Author response image 1.Typical good quality electron density as seen in the β-sheet of VtrA.Shown is the 2mF_o_-DF_c_ map (this is a σ_A_-map, where m is the figure of merit, and D is the model error and scale between F_o_ and F_c_) (Afonine PV, Grosse-Kunstleve RW, Echols N, Headd JJ, Moriarty NW, Mustyakimov M, Terwilliger TC, Urzhumtsev A, Zwart PH, Adams PD. (2012) Towards automated crystallographic structure refinement with phenix.refine. *Acta Crystallogr D Biol Crystallogr*
**68**:352-367.), from the final round of refinement in *PHENIX* contoured at the 1.0 σ level.**DOI:**
http://dx.doi.org/10.7554/eLife.15718.021

Author response image 2.Typical poor density as seen at the solvent-exposed surface of VtrA.Shown is the 2mF_o_-DF_c_ map from the final round of refinement in *PHENIX* contoured at the 1.0 σ level.**DOI:**
http://dx.doi.org/10.7554/eLife.15718.022

A mistake in the space group could end up being worrisome in giving parts of the model wrong (even on top of parts that are right). The new structure in complex with TDC, is in part reassuring (a confirmation with different packing), but some figures of its own, also raise questions.

The authors are acutely aware that F432 is a rather unusual space group, and spent much time and effort to rule out other possibilities. There is no twinning possible in the space group F432, but it is possible for lower symmetry space groups. A summary of the extensive analysis is now presented.

Scaling of data was performed also in the space group F23, a subgroup of F432, to test for the presence of twinning. If the space group is really F23, it could be merohedrally twinned and resemble F432. However, multiple tests indicate that in fact the data as scaled in F23 is NOT twinned tests (for more details on the nature of these tests, please see Padilla JE, Yeates TO. (2003) A statistic for local intensity differences: robustness to anisotropy and pseudo-centering and utility for detecting twinning. *Acta Crystallogr D Biol Crystallogr*
**59**:1124-30; Yeates TO. (1997) Detecting and overcoming crystal twinning. *Methods Enzymol*
**276**:344-58; Yeates TO (1988). Simple statistics for intensity data from twinned specimens. *Acta Crystallogr A***44**:142-4.) Here are the statistics as reported in the Xtriage routine of PHENIX:

a) <I^2^>/<I>^2^ = 2.113 (untwinned: 2.0, perfect twin: 1.5)

b) <F>^2^/<F^2^> = 0.771 (untwinned: 0.785, perfect twin 0.885)

c) <|E^2^-1|> = 0.755 (untwinned: 0.736, perfect twin 0.541)

d) <|L|> = 0.505 (untwinned: 0.500, perfect twin 0.375)

e) <L^2^> = 0.343 (untwinned: 0.333, perfect twin 0.200)

f) In addition, the output from *PHENIX* indicated that the point group of the lattice is F432, and explicitly stated there was no evidence that the lattice displays pseudosymmetry.

Pseudosymmetry typically manifests as systematically weak classes of reflections (for example, h + k = 2n reflections that are systematically weak) (Zwart P.H., Grosse-Kunstleve R. W., Lebedev A. A., Murshudov G. N., Adams P. D. (2008) Surprises and pitfalls arising from (pseudo)symmetry. *Acta Crystallogr D Biol Crystallogr*
**64**:99-107; Li W., Ma C., Guan R., Xu Y., Tomchick D.R., Rizo J. (2011) The crystal structure of a Munc13 C-terminal module exhibits a remarkable similarity to vesicle tethering factors. Structure, **19**:1443-1455). When the data is scaled and merged in *HKL3000* in Laue class m3 (i.e., space group F23), there is no sign of pseudosymmetry in the reflections (i.e., no systematically weak classes of reflections are observed), and the discrepancy indices (i.e., the “R-factors”) are virtually indistinguishable from those obtained when the data is scaled and merged in Laue class m3m (i.e., space group F432). Thus we conclude that the correct space group is in fact F432.

*Issues that have remained unanswered:*

*1) Phasing FOM too low: 0.25 is suspicious for a MAD experiment. Could be explained because of too little anomalous signal (e.g. only two methionines for approx. 200 residues – probably a quite tough phasing scenario). Again, my point was to think (and test!) whether there is a lower symmetry possibility to integrate the data in, and then see if shelxd works better.*

The authors have extensive experience in SAD/MAD phasing of SeMet-substituted soluble proteins, especially with projects that have low anomalous signal and significant radiation decay of the signal. This particular phasing experiment utilized 1 Se site for 108 amino acids (as density for only 216 amino acids was observed and modeled by the final rounds of refinement), hence the chosen phasing method was MAD and not SAD. Data processing and phasing was performed in the graphical user interface of *HKL3000,* with autocorrections implemented to correct for systematic errors such as radiation damage and absorption, and to apply *B*-factor sharpening (Borek D, Dauter Z, Otwinowski Z. (2013) Identification of patterns in diffraction intensities affected by radiation exposure. *J Synchrotron Radiat* 20:37-48.). We have found the multiple corrections applied in the autocorrections routine of *HKL3000* to be critical to our success in phasing numerous difficult anomalous phasing projects. In this case, only the first wavelength (anomalous peak) was chosen for the heavy atom search, as more extensive radiation damage had occurred during the data collection for the second wavelength (inflection point). We performed a search for heavy atom positions to a resolution of 4.0 Å with *SHELXD* which identified two Se positions with correlation coefficients: CC_All_ = 52.9% and CC_Weak_ = 31.2%. It is difficult for the authors to understand what else they could have done with their data that could have resulted in *SHELXD* being able to locate additional selenium sites. Density for only two methionines was observed in the map from the final rounds of model refinement for the native protein (due to disorder of the N-terminal methionines from both monomers of the heterodimer, and the remaining missing methionine is located in the short disordered loop of five residues in VtrC). Both wavelengths were used for phasing in *MLPHARE*, with a resulting FOM of 0.25 for the complete resolution range from 35.7 – 2.60 Å. This compares well with the estimate of anomalous signal calculated in *PHENIX*:

Author response image 3.**DOI:**
http://dx.doi.org/10.7554/eLife.15718.023

In this table, the FOM estimate if the sub-structure (i.e., the Se sites) is solved is 0.3 (+/- 0.1). The P(Substr) is the estimate of probability that the sub-structure can be found with the heavy atom routine *HYSS* using the anomalous peak data (Terwilliger TC, Bunkóczi G, Hung L-W, Zwart PH, Smith JL, Akey DL, Adams PD (20160 Can I solve my structure by SAD phasing? Anomalous signal in SAD phasing. *Acta Crystallogr D Biol Crystallogr*
**72**:346-358).

The FOM after density modification in the program *Parrot* was 0.88, which was expected given that the solvent content of the lattice is 68%. The resulting map was sufficient for an automatic model building routine (*Buccaneer*) to accurately trace and assign sequence to 200 of 238 possible amino acids, or to 93% of all residues in the final model. Thus we believe that this model and space group are correct.

*Straightforward tests that you can also perform to scrutinize this SG assignment: a) use ccp4 program zanuda to analyze your cubic data, it will systematically test for lower compatible space group choices (and also perform some initial refinement for each, to have comparative figures); b) reintegrate the cubic data in P1, and perform mol replacement now that you have a solution (which is maybe completely right, or at least partially right for sure), searching for as many hetero-dimers as you calculate according to the triclinic unit cell volume: if a solution is found, than you can analyze if the packing indeed conforms to a such highly symmetric space group as F432.*

The authors performed this test, and a screenshot of the log file is shown below. The space group F432 appears to be the correct one (*Zanuda* reports larger *R*-free values than reported for the final round of model refinement, as it makes no *TLS B*-factor corrections to the model).

Author response image 4.**DOI:**
http://dx.doi.org/10.7554/eLife.15718.024

2) Maybe it is cubic, end of story.

This suggestion is redundant to the analysis performed in *Zanuda* and thus the authors have not performed this test.

*3) But, if alternative choices do appear, not only shelxd might give better FOM, but actually maps may turn out better (in the bad parts, if there were); final refined average B factors closer to Wilson B (this is a definitely strange behavior reported in your table); better Rmerge statistics (also high, not only in the high resolution shell, but also in the low res – I know high multiplicity is here at play, so I do not want to say this is necessarily wrong: just look closely and critically at it, to avoid mistakes).*

An extensive discussion with one of the authors of *HKL3000* (Zbyszek Otwinowski) reveals the nature of the confusion regarding the Wilson *B*-factor. The data was scaled in *HKL3000* using the autocorrections routine, which performs extensive *B*-factor sharpening; only the sharpened *B*-factors were originally reported. The unsharpened Wilson *B*-factors for all datasets are now included in Table 1. For the SeMet and native F432 structures, the unsharpened Wilson *B*-factors are 77.6 to 79.0, very similar with the observed mean *B*-values for the refined model.

It is a well-known consequence of high multiplicity (in the case of the native model, 36-fold) that the values for *Rmerge* will be artificially higher than expected, even though the merged data are improved by averaging more observations. This is the reason that reported in Table 1 are values for *Rpim*, the precision indicating *R*-factor (Evans P.R. (2011) An introduction to data reduction: space group determination, scaling and intensity statistics. *Acta Crystallogr D Biol Crystallogr*
**67**:282-92.), which estimates the data quality after merging. These values are roughly a factor of 6-fold lower than the *Rmerge* values, and we think these values more accurately reflect the quality of the native data set. In addition, the CC_1/2_ value for the highest resolution shell (0.934) generously exceeds that recommended as a suitable high-resolution cutoff limit (Diederichs K. and Karplus P.A. (2012) Linking crystallographic model and data quality. *Science* 336:1030-1033).

*4) I would strongly advise against including hydrogen atoms in the refinements. Else, explain really well why you think it's OK to include them. This goes also for the orthorhombic form with TDC. The validation reports seem clear in that you have included them: to support a refinement strategy with independent H atoms you would need ultra-high resolution (i.e. better than 1.2Å) or a neutron light source for the diffraction experiment.*

Modern methods of macromolecular crystallographic refinement encourage the inclusion of hydrogen atoms at ALL resolutions. These hydrogen atoms are not located independently in an electron density map and refined independently of the non-hydrogen atoms in the model; rather, they are placed in geometrically idealized conformations with relation to the non-hydrogen atoms they are covalently bonded to and during positional refinement they are tied to their bonded atoms. These are thus known as “riding hydrogen atoms” and they have been used for decades in small-molecule refinements (especially in cases where hydrogen atoms are not explicitly observed in the electron density maps). Since 2010 the program authors of *PHENIX* have encouraged users to incorporate riding hydrogen atoms into their refinement protocols (Adams P.D., *et al.* (2010) PHENIX – a comprehensive Python-based system for macromolecular structure solution. *Acta Crystallographica D Biol Crystallogr*
**66**:213–221; Afonine PV, Mustyakimov M, Grosse-Kunstleve RW, Moriarty NW, Langan P, Adams PD (2010) Joint X-ray and neutron refinement with phenix.refine. *Acta Crystallogr D Biol Crystallogr*
**66**:1153-63.), and in fact, *REFMAC* adds the riding hydrogen atoms by default (http://www.ccp4.ac.uk/html/refmac5/keywords/restraints.html). The reasons for including riding hydrogen atoms in refinement are two-fold: 1) the hydrogen atoms actually exist in the molecules and contribute to the X-ray scattering at low resolution, so their inclusion in the refinement improves the agreement between *F_obs_*and *F_calc_* at low resolution; and 2) their inclusion improves the geometry of the refined model, as they are used to analyze clashes due to van der Waals repulsions. Jane and David Richardson have shown the utility of this method with the *MolProbity* server, which requires hydrogen atoms for the all-atom contact analysis it performs for validation of macromolecular structures (Chen et al. (2010) MolProbity: all-atom structure validation for macromolecular crystallography. *Acta Crystallographica D Biol Crystallogr*
**66**:12-21.).

The two major refinement packages handle riding hydrogen atoms differently. *REFMAC* utilizes implicit hydrogen atoms and does not automatically write them out to the resulting PDB file (although the user may choose to explicitly to write them out), and *PHENIX* automatically includes them in the resulting PDB file. The user would actively need to remove the hydrogen atoms from a file generated by the *PHENIX* refinement program prior to PDB deposition. The authors of this work have chosen to include all atoms used to calculate the structure factors that are deposited along with the model coordinates in the PDB, thus statistics for the riding hydrogen atoms appear in the PDB validation report.

*5) The nominal resolution for the new orthorhombic structure is probably not 2.1Å: very low completeness of the data (57% high res shell). Please reconsider this. Actually, please elaborate as to why completeness drops significantly comparing the processed data vs those used for refinement.*

This dataset displays significant anisotropy, which results in the relatively low completeness in the highest resolution shell during data collection (79.4%). Anisotropic diffraction was corrected to adjust the error model and to compensate for a radiation-induced increase of non-isomorphism within the crystal. These corrections were applied to the data during scaling in the autocorrections routine of *HKL3000* and thus are responsible for the difference in completeness between the data collection and refinement (Borek D, Dauter Z, Otwinowski Z. (2012) Identification of patterns in diffraction intensities affected by radiation exposure. *J Synchrotron Radiat*
**20**:37-48; Borek D, Cymborowski M, Machius M, Minor W, Otwinowski Z. (2010) Diffraction data analysis in the presence of radiation damage. *Acta Crystallogr D Biol Crystallogr*
**66**:426-36; Borek D, Ginell SL, Cymborowski M, Minor W, Otwinowski Z. (2007) The many faces of radiation-induced changes. *J Synchrotron Radiat*
**14**:24-33.).

The resolution chosen as the refinement limit (2.10 Å) was chosen carefully and is consistent with current practices.

a) The I/σ(I) is greater than 1.0 and less than 3.0;

b) The percentage completeness in the highest resolution shell is greater than 50%;

c) The CC_1/2_ in the highest resolution shell is 0.754, and greater than 0.5 (this value for all datasets has been included in Table 1;

d) The multiplicity in the highest resolution shell is greater than 2.0;

e) The electron density map, the refinement statistics and the model stereochemical parameters degrade in quality when the highest resolution limit used for refinement is 2.15 Å.

It is current practice to include data from resolution shells that in the past were ignored due to a high percentage of weak data and/or lower than 100% completeness due to anisotropy (Diederichs K. and Karplus P.A. (2012) Linking crystallographic model and data quality. *Science*
**336**:1030-1033; Weiss M.S. (2001) Global indicators of X-ray data quality. *J Appl Cryst*
**34**:130-135; Diederichs K. and Karplus P.A. (1997) Improved *R*-factors for diffraction data analysis in macromolecular crystallography. *Nat Struct Mol Biol*
**4**:269-274.). In this case, inclusion of the data in resolution shells to 2.10 Å actually improves the model and refinement statistics, which are excellent, and thus the authors have not changed the upper resolution limit for the structure with the bile salt.

*It is suggested to reduce/simplify subsection “The VtrA/VtrC complex is an obligate heterodimer”; it's enough there to indicate that only 2 Se-Met out of the 5 present in the protein complex, were ultimately seen to be ordered in the refined model, hence contributing proportionally little to the measured anomalous signal in the first place (likely explaining the low FOM figure at the phasing step, previous to density modification).*

In this context it is rather misleading to state that the 2 Se-Met "were used for refinement", actually you are probably talking here about their use to solve the substructure of anomalous scatterers for subsequent protein phasing purposes.

Indeed the wording that should have been used in this case was “Two of the five methionines, VtrA Se-Met 210 at the interface with VtrC and VtrC Se-Met 49, were used for phasing.” This sentence, plus the following two sentences, were added in response to the previous reviewer questions regarding how the structure could have been phased with only two selenomethionine sites. Thus we will replace these three sentences with the following shorter, clearer sentences:

“Two of the five selenomethionines were used for phasing. The two N-terminal methionines and Se-Met 115 in the loop that covers the β barrel of VtrC were not observed in the final electron density map and were not included in the final model.”